# NeurOCNN: A Neural-Operator-Based Model for Physiological Time Series

**Daya Kumar** [1 2] **Uday Devulapalli** [1 2] **Aarat Satsangi** [1 2] **Apurva Narayan** [1]

## Abstract

Neural operators have become a central tool in scientific machine learning for learning discretization-consistent solution operators, achieving strong results on partial differential equation (PDE) benchmarks. Physiological time series, however, are highly nonstationary and dominated by localized transient events, properties that can challenge both PDE-oriented neural operators and conventional deep models. We propose NeurOCNN, a neural-operator-based model for physiological signals that learns a robust function-to-label mapping. NeurOCNN integrates continuous-time, spline-parameterized convolutions to capture localized morphology with Fourier projection pooling for variable-to-fixed dimensional mapping, thereby enabling robust, discretization-invariant inference under zero-shot sampling-rate shifts. Empirically, NeurOCNN outperforms standard neural-operator baselines, achieves performance comparable to state-of-the-art methods, and maintains stable accuracy across multiple previously unseen evaluation sampling rates. The code is available at https://github.com/Idsl-group/NeurOCNN.

## 1. Introduction

Neural operators have emerged as a compelling paradigm for learning discretization-consistent mappings between functions, with notable successes in scientific computing and surrogate modeling for Partial Differential Equations (PDEs) (Kovachki et al., 2023). Architectures such as Fourier Neural Operators (FNOs) (Li et al., 2021) and Deep-ONet (Lu et al., 2021) parameterize operators in ways that can generalize across resolutions, enabling models trained on one discretization to be deployed on others without re-

[1]University of Western Ontario, London, Canada [2]International Center for Applied Systems Science for Sustainable Development (ICASSSD), Cambridge, Canada. Correspondence to: Apurva Narayan <apurva.narayan@uwo.ca>.

*Proceedings of the 43rd International Conference on Machine Learning*, Seoul, South Korea. PMLR 306, 2026. Copyright 2026 by the author(s).

designing the architecture. This discretization robustness is valuable whenever the underlying phenomenon is continuous but only observed as sampled data (Bleistein & Guilloux, 2024), motivating operator-learning approaches that generalize across resolutions.

Physiological time series provide a motivating yet challenging domain for operator learning. Signals such as electroencephalography (EEG), electrooculography (EOG), and electrocardiography (ECG) exhibit rich non-stationarity, heterogeneous acquisition protocols, and event-like localized temporal patterns (Reilly & Lee, 2010). Practical prediction tasks span a range of objectives, including sleep staging (Eldele et al., 2021; Perslev et al., 2021), arrhythmia detection (Hannun et al., 2019), movement or arousal characterization (Atzori et al., 2014; Ghassemi et al., 2018), and clinical risk scoring (Ribeiro et al., 2020). While modern deep learning has improved task performance substantially (Supratak et al., 2017; Hannun et al., 2019), many models remain coupled to particular discretizations and preprocessing choices, such as fixed sampling rates, fixed-length tokenizations, and modality-specific pipelines (Supratak et al., 2017; Nie et al., 2023; Jiang et al., 2024). This can complicate reuse across datasets and settings where the same underlying physiology is observed at different temporal resolutions.

A natural response is to cast physiological signal prediction in an operator-learning framework. Let a multichannel physiological segment be a function $\mathbf{x} : [0, T] \to \mathbb{R}^C$, where $t \in [0, T]$ denotes physical time (seconds), $C$ is the number of channels, and $\mathbf{x}(t) \in \mathbb{R}^C$ gives channel amplitudes at time $t$. The goal is to predict a task target $y \in \mathcal{Y}$. In this view, the learning problem is to approximate a mapping $\mathcal{G} : \mathbf{x}(\cdot) \mapsto y$ that is defined over continuous-time functions rather than tied to a particular sampled grid.

However, neural operators developed for PDEs do not readily extend to physiological time series. Many canonical neural operators are designed for *function-to-function* mappings (Kovachki et al., 2023) and often emphasize global spectral representations that are well aligned with smooth solution operators (Li et al., 2021). In contrast, physiological signals frequently contain abrupt transients, localized morphology changes, and structured artifacts (e.g., motion, electrode drift, muscle activity) that can be poorly captured by purely global parameterizations (Liu-Schiaffini et al.,

2024b). Moreover, physiological prediction commonly requires *function-to-vector* outputs under substantial inter-subject variability and noise, which shifts the modeling emphasis toward robust feature extraction and task-aware aggregation while still benefiting from discretization consistency.

In parallel, large-scale sequence models and foundation models for physiological signals have advanced representation learning by leveraging pretraining and scale (Jiang et al., 2025; 2024; Wang et al., 2024; McKeen et al., 2025; Kurbis et al., 2025). These approaches can transfer effectively across tasks and datasets, yet they remain fundamentally finite-grid learners: they operate on discretized sequences and typically assume a fixed temporal representation at training and inference. As a result, real-world pipelines often rely on resampling or interpolation as compatibility steps, implicitly treating discretization as a modeling prerequisite rather than an incidental measurement choice. This motivates architectures whose computations are defined with respect to physical time, so that changes in sampling rate do not necessitate redesigning tokenizations or retraining for each acquisition configuration.

We propose **NeurOCNN**, a neural-operator-based model for physiological time series that instantiates $\mathcal{G}$ using three components: (i) a *continuous-time* convolutional operator with a spline-parameterized kernel (SplineConv1d), (ii) a Fourier projection pooling module that maps variable-resolution signals into a fixed-dimensional representation, and (iii) an attention-based task head designed to handle heterogeneous downstream tasks. This architecture explicitly targets the local, nonstationary structure of physiological signals while retaining the discretization robustness that motivates operator learning. Importantly, robustness to sampling-rate shifts follows naturally from the continuous-time formulation and time-referenced pooling, rather than being imposed as a separate constraint. Empirically, across multiple datasets spanning EEG, EOG and ECG, NeurOCNN outperforms standard neural operator baselines and achieves performance comparable to strong domain models, while maintaining stable accuracy under unseen sampling rates.

## Contributions

- **Operator viewpoint for physiological prediction.** We formalize physiological time-series decoding as learning a function-to-label operator $\mathcal{G} : \mathbf{x}(\cdot) \mapsto y$ from continuous-time multichannel signal functions to task targets.

- **NeurOCNN: an operator architecture for physiological signals.** We introduce a neural-operator-based model that combines a continuous-time spline-parameterized convolutional operator with Fourier pro-

jection pooling and an attention-based task head, designed to capture localized and nonstationary patterns common in EEG, EOG and ECG.

- **Benchmarking against operator and domain baselines.** We evaluate NeurOCNN against established neural operator families and strong modality-specific baselines across multiple physiological datasets. Our results demonstrate that NeurOCNN outperforms generic operator adaptations and matches state-of-the-art accuracy while remaining robust to changes in sampling frequency at inference.

- **Sampling-rate robustness.** By grounding computations in physical time and using Fourier basis projection pooling, NeurOCNN exhibits stable inference under sampling-rate shifts without requiring resampling-based standardization.

## 2. Related Work

**Neural operators.** Neural operators extend conventional neural networks from finite-dimensional input-output mappings to *maps between function spaces*, with the central aim of *discretization-consistent* learning: a model trained on one sampling grid can be evaluated on another without changing parameters (Kovachki et al., 2023; Li et al., 2021; Lu et al., 2021). Canonical instances such as the FNO achieve this via global spectral mixing (Li et al., 2021), while DeepONet provides an alternative operator approximation construction via a branch-trunk decomposition (Lu et al., 2021). Subsequent work has improved expressivity and multiscale modeling through architectural refinements and alternative bases, including U-shaped operator backbones (U-NO) with skip connections (Rahman et al., 2023), localized multiwavelet representations (Gupta et al., 2021), and attention-based operator learners with Fourier/Galerkin-inspired formulations (Cao, 2021). Related directions include spectral-series operator parameterizations (Spectral Neural Operator (SNO) (Fanaskov & Oseledets, 2023)) and explicitly localized operator constructions (Local Neural Operator (LocalNO) (Liu-Schiaffini et al., 2024a)). Despite these advances, the dominant evaluation setting remains PDE-oriented *function-to-function* prediction, where outputs are fields and global mixing is often well matched to smooth solution operators (Kovachki et al., 2023; Li et al., 2021).

Physiological decoding differs in two key ways: targets are commonly *function-to-target*, and discriminative evidence can be highly localized and nonstationary (e.g., transient morphology changes and artifacts), motivating operator instantiations that emphasize time-local feature extraction and task-aware aggregation rather than relying primarily on global spectral parameterizations.

**Physiological foundation models and continuous-time sequence learning.** In parallel, physiological signal modeling has advanced rapidly through self-supervised and foundation-model approaches that learn transferable representations from large collections of discretized recordings, typically using contrastive or masked-reconstruction objectives (Kiyasseh et al., 2021; Jiang et al., 2024; Wang et al., 2024). These methods can yield strong transfer and data efficiency, but their core computation is still defined on sampled sequences and depends on tokenization and preprocessing choices that implicitly fix a temporal discretization.

Separately, continuous-time sequence models such as Neural Ordinary Differential Equations (Neural ODEs) and Neural Controlled Differential Equations (Neural CDEs) define latent dynamics in continuous time to process irregular sampling (Chen et al., 2018; Kidger et al., 2020). However, these models are typically structured as end-to-end sequence learners rather than modular, operator-style decoders optimized for robust, cross-resolution classification interfaces. While continuous-kernel convolutional models parameterize filters as continuous functions to accommodate variable resolution, standard formulations parameterized by unconstrained MLPs retain excessive architectural capacity (Romero et al., 2022; Knigge et al., 2023). This flexibility allows them to overfit to discretization-specific artifacts on the native training grid, rendering their performance highly sampling-rate dependent when evaluated under mismatched test resolutions (see A.2.1). NeurOCNN addresses these limitations by combining a continuous-time, spline-parameterized convolutional operator, which enforces an implicit spectral regularization, with fixed-dimensional Fourier projection pooling and a dedicated task head, enabling stable, discretization-consistent decoding across unseen sampling rates.

## 3. Methodology

### 3.1. Problem Formulation

We study supervised decoding from physiological time series observed over a finite horizon $[0, T]$. Each example is modeled as a multichannel continuous-time signal

$$x : [0, T] \to \mathbb{R}^{C_{\text{in}}}, \tag{1}$$

and the goal is to learn a *function-to-label* operator

$$\mathcal{G}^\star : \mathcal{X} \to \{1, \dots, C_{\text{cls}}\}, \qquad \hat{y} = \mathcal{G}_\theta(x), \tag{2}$$

where $\mathcal{X}$ denotes an appropriate function space over $[0, T]$.

We observe $x$ through uniform sampling at a device-dependent rate $f_s$ (Hz):

$$t_n = \frac{n}{f_s}, \quad n = 0, \dots, N-1, \qquad N = \lfloor T f_s \rfloor, \tag{3}$$

yielding a discrete tensor $X \in \mathbb{R}^{C_{\text{in}} \times N}$ with $X_{c,n} = x_c(t_n)$ (and a batch $X \in \mathbb{R}^{B \times C_{\text{in}} \times N}$).

Learning an operator in Eq. (2) means that $\mathcal{G}_\theta$ is defined on the underlying continuous-time signal and can therefore be evaluated on different discretizations of the same function, equivalently, the input to the model may be provided on arbitrary uniform sampling grids (with varying $f_s$) without changing the learned parameters, aside from the numerical approximation induced by discretizing continuous-time computations.

### 3.2. Architecture Overview

Our predictor factorizes into a continuous-time operator backbone, a projection-based pooling stage, and a task head:

$$\hat{y} = \mathcal{H}_\psi \Big( \mathcal{P}\Big( \mathcal{B}_\theta(X; f_s), \, f_s \Big) \Big). \tag{4}$$

Here, $\mathcal{B}_\theta$ produces a time-indexed feature sequence, $\mathcal{P}$ maps this variable-length sequence to a fixed set of tokens via Fourier projection, and $\mathcal{H}_\psi$ maps tokens to outputs using attention and an MLP.

### 3.3. Continuous-Time Spline Convolution Operator

A multi-channel continuous-time convolution can be written as the integral operator

$$(\mathcal{K}x)_o(t) = \sum_{i=1}^{C_{\text{in}}} \int_{\mathbb{R}} k_{o,i}(\tau) \, x_i(t - \tau) \, d\tau, \tag{5}$$

where $k_{o,i} : \mathbb{R} \to \mathbb{R}$ is a learnable kernel function, $i$ indexes input channels, and $o$ indexes output channels. To avoid learning discrete filters tied to a particular sampling rate, we represent each kernel $k_{o,i}(\tau)$ as a smooth function over a compact support of duration $D$ seconds using a (natural) cubic spline (Fey et al., 2018).

Let $\{\xi_j\}_{j=1}^P \subset [-\frac{1}{2}, \frac{1}{2}]$ be fixed knot locations (uniformly spaced), and let $\theta_{o,i} \in \mathbb{R}^P$ denote learnable control values for channel pair $(o, i)$. We define $k_{o,i}(\tau)$ by evaluating the spline induced by $(\xi_j, \theta_{o,i,j})$ at normalized time $\xi = \tau/D \in [-\frac{1}{2}, \frac{1}{2}]$.

**Sampling the kernel at rate $f_s$.** Given a sampling rate $f_s$, we discretize the kernel by choosing

$$L = \max\{1, \text{round}(D f_s)\}, \tag{6}$$

and enforcing an odd length by incrementing $L$ if it is even. Let $\{u_\ell\}_{\ell=1}^L$ be a uniform grid on $[-\frac{1}{2}, \frac{1}{2}]$. We form an interpolation matrix $A \in \mathbb{R}^{L \times P}$ that maps control values to sampled taps, where $A_{\ell,j}$ is the value of the $j$-th spline basis function at $u_\ell$. (Implementation detail: $A$ is obtained by evaluating a natural cubic spline fit to the basis vectors and is cached per $(L, \text{dtype})$.) The sampled taps are

$$w_{o,i,\ell} = \sum_{j=1}^P \theta_{o,i,j} \, A_{\ell,j}, \qquad \ell = 1, \dots, L. \tag{7}$$

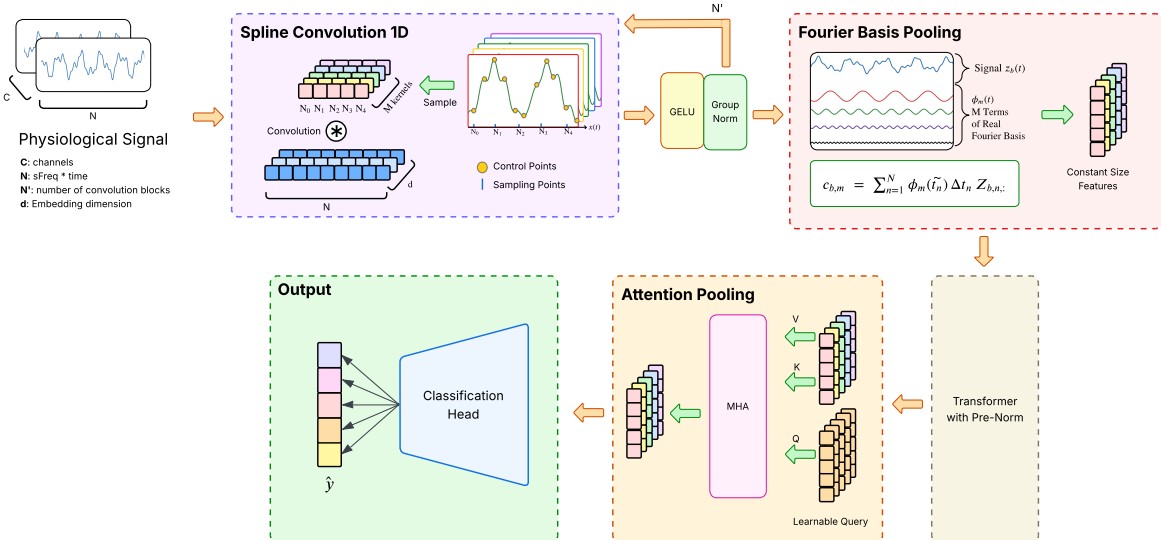

*Figure 1.* Overview of the proposed architecture. First, the bio-signal passes through a feature extractor consisting of $N'$ repeating spline-based convolution blocks with GELU activations and group normalization. Next, Fourier basis pooling converts the signal into fixed-size features. A Transformer layer is then applied for token mixing, followed by attention pooling using a learnable query parameter. Finally, the representation is passed to a classification head.

**Riemann-sum scaling and discrete convolution.** For samples $X_{b,i,n} \approx x_{b,i}(t_n)$, we compute a discrete convolution with taps $w$ and multiply by $\Delta t$ to obtain a consistent Riemann-sum approximation of Eq. (5):

$$Z_{b,o,n} = \Delta t \sum_{i=1}^{C_{\text{in}}} \sum_{\ell=0}^{L-1} w_{o,i,\ell} \, \widetilde{X}_{b,i,n+\ell-\lfloor L/2 \rfloor}, \qquad (8)$$

where $\widetilde{X}$ denotes the symmetrically padded input used for "same" convolution. The convolution stride can be specified in seconds as $s_{\text{sec}}$; at sampling rate $f_s$ the integer stride is $s = \max\{1, \text{round}(s_{\text{sec}} f_s)\}$, and the effective output sampling rate becomes $f_s^{\text{out}} = f_s/s$. In our NeurOCNN configuration, spline convolutions use unit stride ($s = 1$), so $f_s^{\text{out}} = f_s$.

**Backbone instantiation.** We stack four spline-convolution layers with channel widths $C_{\text{in}} \rightarrow 64 \rightarrow 128 \rightarrow 64 \rightarrow 32$. After each of the first three convolutions we apply GELU and group normalization (one group); after the fourth convolution we apply group normalization. This yields time-indexed features which we reorder to time-major tokens $Z \in \mathbb{R}^{B \times N \times d}$ with embedding dimension $d = 32$.

### 3.4. Fourier Basis Projection Pooling

The backbone produces a time-indexed feature sequence $Z \in \mathbb{R}^{B \times N \times d}$ whose length $N$ depends on the sampling rate $f_s$. We obtain a fixed-length representation by projecting the features onto a truncated Fourier basis using a quadrature-weighted inner product ([Rippel et al., 2015](); [Yao]()

et al., [2021]()).

**Fourier basis projection.** For each sample $b$, view $Z_{b,n,:}$ as samples of a continuous-time embedding $z_b : [0, T] \rightarrow \mathbb{R}^d$ at timestamps $\{t_n\}_{n=1}^N$ (in seconds). Given a horizon $\widetilde{T} > 0$, define a real Fourier basis on $[0, \widetilde{T}]$ with $M$ terms:

$$\omega_k := \frac{2\pi k}{\widetilde{T}}, \qquad k \geq 1, \qquad (9)$$

$$\phi_0(t) = 1, \quad \phi_{2k-1}(t) = \sin(\omega_k t), \quad \phi_{2k}(t) = \cos(\omega_k t). \qquad (10)$$

truncated to $m = 0, \ldots, M - 1$. We compute coefficient (token) vectors by integrating features against each basis function:

$$c_{b,m} = \int_0^{\widetilde{T}} \phi_m(t) \, z_b(t) \, dt \in \mathbb{R}^d, \quad m = 0, \ldots, M - 1. \qquad (11)$$

On the observed grid, we use a quadrature-weighted sum with per-sample time steps $\Delta t_n$:

$$c_{b,m} \approx \sum_{n=1}^{N} \phi_m(\tilde{t}_n) \, \Delta t_n \, Z_{b,n,:}, \qquad (12)$$

where $\tilde{t}_n$ denotes timestamps rebased to start at zero and clamped to $[0, \widetilde{T}]$. We set $\Delta t_n = t_{n+1} - t_n$ and repeat/clamp the final step for numerical stability. Stacking $\{c_{b,m}\}_{m=0}^{M-1}$ yields $C_b \in \mathbb{R}^{M \times d}$, i.e., $M$ tokens per example. Because Eq. (12) uses time weights in seconds, the coefficients approximate the same continuous-time integrals across different sampling rates.

**Segment-wise Fourier basis projection pooling.** In a Fourier basis on an interval of length $\widetilde{T}$, harmonic index corresponds to *physical frequency*: the $k$-th harmonic represents $f_k = k/\widetilde{T}$ Hz. With $M$ real-basis terms, the maximum represented frequency scales as

$$f_{\max} \approx \frac{(M-1)}{2\widetilde{T}}. \tag{13}$$

If we pool over a longer time horizon (larger $\widetilde{T}$), preserving coverage up to a target frequency requires increasing $M$ roughly linearly with $\widetilde{T}$. Moreover, physiological events are often nonstationary and localized; representing localized structure using basis functions spanning a long interval can require many coefficients. To keep $M$ moderate while maintaining frequency coverage and locality, we apply the projection within short segments.

We partition the window $[0, T]$ into $S$ contiguous segments of length $T_{\text{seg}}$ such that $T = ST_{\text{seg}}$:

$$I_s = [(s-1)T_{\text{seg}}, sT_{\text{seg}}], \qquad s = 1, \ldots, S. \tag{14}$$

For each segment $s$, we extract the subsequence $Z_{b,s} \in \mathbb{R}^{N_s \times d}$ whose timestamps fall in $I_s$, rebase time to $[0, T_{\text{seg}}]$ by subtracting the segment start (and clamping), and apply the same quadrature projection as Eq. (12) with $\widetilde{T} = T_{\text{seg}}$:

$$C_{b,s}[m,:] = \sum_{n=1}^{N_s} \phi_m^{(T_{\text{seg}})}(t_{s,n}) \, \Delta t_{s,n} \, Z_{b,s}[n,:] \in \mathbb{R}^d, \\ m = 0, \ldots, M-1. \tag{15}$$

We treat each row $C_{b,s}[m,:]$ as a token and concatenate across segments:

$$U_b \in \mathbb{R}^{K \times d}, \qquad K = SM, \tag{16}$$

yielding $U \in \mathbb{R}^{B \times K \times d}$. Segmenting reduces the projection horizon from $T$ to $T_{\text{seg}}$, increasing the effective frequency coverage for fixed $M$:

$$f_{\max}^{\text{seg}} \approx \frac{(M-1)}{2T_{\text{seg}}}. \tag{17}$$

Equivalently, the overall tokenization corresponds to projecting onto a simple windowed Fourier family $\psi_{s,m}(t) = \mathbf{1}_{I_s}(t) \, \phi_m^{(T_{\text{seg}})}(t - (s-1)T_{\text{seg}})$.

### 3.5. Token Mixing and Prediction Head

Given the Fourier projection tokens $U \in \mathbb{R}^{B \times K \times d}$, we first apply token mixing using multi-head self-attention, followed by a Transformer encoder stack:

$$\widetilde{U} = \text{MHA}(U, U, U), \qquad V = \text{Enc}(\widetilde{U}), \tag{18}$$

where MHA denotes multi-head attention, $K$ is the number of Fourier tokens, and $d$ is the embedding dimension.

To aggregate the token sequence into a fixed-dimensional representation, we use learnable-query attention pooling. Specifically, we learn a query token $q \in \mathbb{R}^{1 \times d}$ and use it to attend to the encoded token sequence:

$$\bar{z}_b = \text{MHA}(q, V_b, V_b), \qquad z_b = \text{LN}(\bar{z}_b), \tag{19}$$

where LN denotes layer normalization, $V_b \in \mathbb{R}^{K \times d}$ denotes the encoded tokens for sample $b$, and $z_b \in \mathbb{R}^d$ is the pooled representation. Finally, an MLP maps this representation to classification logits:

$$\hat{y}_b = \text{MLP}(z_b), \qquad \hat{y}_b \in \mathbb{R}^{C_{\text{cls}}}. \tag{20}$$

### 3.6. Datasets

Sleep is a high-impact public-health domain as insufficient sleep and sleep disorders are both common and consequential. Recent reports have indicated that the share of U.S. adults with short sleep ($< 7$ hours in a 24-hour period) remains high, while NHLBI estimates that 50 to 70 million Americans live with sleep disorders and that about one in three adults do not regularly obtain the recommended amount of uninterrupted sleep (Centers for Disease Control and Prevention, 2024; National Heart, Lung, and Blood Institute, 2023). These realities motivate learning methods that can extract clinically meaningful information from physiological recordings across diverse cohorts and acquisition conditions. Sleep staging, in particular, is a standardized, expert-labeled clinical task that provides a rigorous testbed for evaluating robustness to real-world variability. Given this, we use multiple complementary sleep benchmarks: Sleep-EDF (Kemp et al., 2000; Goldberger et al., 2000), ISRUC (Khalighi et al., 2016), and HMC (Alvarez-Estevez & Rijsman, 2022; 2021) (community and clinical polysomnography with expert sleep-stage labels) and Sleep-EOG (Kemp et al., 2000) (a reduced-modality EOG-only benchmark derived from Sleep-EDF), and we additionally include ECG rhythm classification (Clifford et al., 2017) to demonstrate that the proposed model extends beyond sleep to other physiological time-series tasks. Dataset descriptions and preprocessing steps are provided in Appendix A.1.

### 3.7. Baselines

We compare NeurOCNN against two groups of methods. First, to isolate the effect of the operator backbone under a controlled tokenization and head, we evaluate four neural-operator backbones - FNO (Li et al., 2021), U-NO (Rahman et al., 2023), SNO (Fanaskov & Oseledets, 2023), and LocalNO (Liu-Schiaffini et al., 2024a) - each paired with the same *Fourier projection pooling* module used in our model. This standardizes the interface to a fixed-size token sequence across sampling rates and ensures differences are primarily attributable to the backbone operator rather than downstream aggregation.

Second, we compare against strong non-operator baselines that are widely used for physiological time-series modeling. As a compact and well-established convolutional baseline for EEG decoding, we include EEGNet (Lawhern et al., 2018). For the sleep-staging benchmarks, we report results for two high-performing task-specific architectures, AttnSleep (Eldele et al., 2021) and U-Sleep (Perslev et al., 2021), which represent strong modern baselines in this setting. To benchmark against large-scale representation learning, we include LaBraM (Jiang et al., 2024), a pretrained EEG foundation model. In addition, we evaluate two further competitive sequence-modeling baselines: a standard ResNet backbone (He et al., 2016) and PatchTST (Nie et al., 2023), a transformer-based architecture designed for time-series classification.

### 3.8. Experimental Setup

We use 5-fold cross-validation for all experiments. For the sleep datasets (Sleep-EDF, SleepEOG, ISRUC, and HMC), subject identifiers are available; therefore, we enforce subject-level separation between the training and test sets in each fold to prevent subject leakage. The folds are stratified where possible to keep the label distribution approximately balanced. For the ECG benchmark, where subject identifiers are not available, we use standard sample-level splits with stratification by label.

To evaluate discretization robustness, we train each model at the dataset's native sampling rate $f_{\text{train}}$ and evaluate zero-shot (without retraining) across both lower (temporal coarsening) and higher (temporal super-resolution) test sampling rates. For each test rate, we resample each input window accordingly, pass the corresponding scalar sampling rate to models that accept $f_s$, and keep all learned parameters fixed. For conventional baselines that require a fixed discretization, we resample the test-rate signal back to $f_{\text{train}}$ before inference. This preserves architectural compatibility while still probing sensitivity to rate-conversion artifacts. The minimum test sampling rate is chosen to be consistent with the preprocessing low-pass filtering to satisfy Nyquist and avoid aliasing.

We report classification accuracy for all tasks. All models are trained with the AdamW optimizer (learning rate $10^{-3}$) and optimized using the cross-entropy loss.

## 4. Results

### 4.1. Comparisons Across Datasets

Table 1 summarizes cross-dataset test accuracy under 5-fold cross-validation, averaged across all evaluation sampling frequencies. Across modalities, NeurOCNN matches the strongest task-specific baselines while consistently outperforming standard neural-operator models paired with the

same Fourier projection pooling. On ISRUC and HMC, NeurOCNN is within 0.46 and 0.98 points of U-Sleep, respectively, but with lower variability (e.g., 1.04 pp vs. 1.50 pp on ISRUC), indicating stronger cross-frequency stability at near–state-of-the-art accuracy. Relative to canonical operator baselines, NeurOCNN delivers large gains (e.g., +4.23 points over the best operator baseline on ISRUC, *LocalNO+FP*, and +3.56 on HMC), suggesting that NeurOCNN's spline-kernel backbone plays a central role in the observed improvements under a common pooling mechanism.

This trend extends beyond EEG. On Sleep-EDF (EOG), NeurOCNN is within 0.43 points of the best method (AttnSleep) while remaining competitive in variability. On ECG, NeurOCNN achieves the best accuracy (76.08%), exceeding ResNet and outperforming operator baselines by a wide margin. Notably, these results are obtained with a compact model (319.59K parameters), reinforcing that NeurOCNN's improvements stem from architectural inductive bias and discretization robustness rather than scale. Full per-frequency test results for all models on all four datasets (i.e., performance at each evaluation sampling rate) are reported in Appendix A.2.2.

### 4.2. Discretization Robustness

Figures 2a, 2b report test accuracy as a function of the test sampling frequency $f \in \mathcal{F}$ on ISRUC and HMC, where each point corresponds to the mean test accuracy across 5 folds. Figure 2c summarizes robustness via the *worst-case accuracy drop per fold* relative to the training sampling frequency $f_{\text{train}}$.

Formally, let $a_k(f)$ denote the test accuracy on fold $k$ when the test set is resampled to frequency $f$. The line plots visualize the fold-averaged accuracy. To quantify discretization robustness, for each fold we compute the accuracy drop relative to $f_{\text{train}}$ and take the worst case across all tested sampling frequencies:

$$\Delta_k = \max_{f \in \mathcal{F}} \big( a_k(f_{\text{train}}) - a_k(f) \big), \qquad (21)$$

$$f_k^\star = \arg\max_{f \in \mathcal{F}} \big( a_k(f_{\text{train}}) - a_k(f) \big), \qquad (22)$$

where $\Delta_k$ is the worst-case drop for fold $k$ and $f_k^\star$ is the corresponding worst-case test frequency (both summarized in Figure 2c).

Across both ISRUC and HMC, NeurOCNN exhibits a comparatively *flat* accuracy-frequency profile, with substantially smaller variation in $\bar{a}(f)$ across the tested frequencies than most baselines. This behavior is reflected in Figure 2c, where NeurOCNN attains among the smallest worst-case drops $\{\Delta_k\}_{k=1}^5$ and shows low fold-to-fold variability, indicating robust generalization under test-time resampling. In

*Table 1.* Test performance across all sampling frequencies over 5-fold cross-validation. Values are mean test accuracy (%) ± standard deviation (in percentage points). Fourier Pooling is denoted by FP. Best accuracy and second-best accuracy are color-coded.

| Model | Params | ISRUC | HMC | Sleep-EDF (EOG) | ECG |
|---|---|---|---|---|---|
| FNO + FP | 5.23 M | 66.48 ± 2.07 | 66.85 ± 1.21 | 48.61 ± 0.83 | 60.31 ± 4.25 |
| U-NO + FP | 1.44 M | 66.69 ± 1.33 | 60.57 ± 7.48 | 46.86 ± 2.31 | 59.42 ± 2.57 |
| SNO + FP | 273.24 K | 61.58 ± 2.75 | 62.32 ± 1.68 | 41.00 ± 1.11 | 58.16 ± 0.07 |
| LocalNO + FP | 57.00 K | 70.99 ± 1.90 | 69.32 ± 1.72 | 54.92 ± 1.97 | 65.83 ± 1.14 |
| EEGNet | 8.58 K | 70.12 ± 3.06 | 67.48 ± 4.27 | – | – |
| AttnSleep | 551.61 K | 72.74 ± 1.83 | 67.64 ± 1.54 | 56.80 ± 1.37 | – |
| U-Sleep | 3.71 M | 75.68 ± 1.50 | 73.86 ± 2.35 | 51.39 ± 2.13 | – |
| LaBraM | 11.61 M | 68.71 ± 1.63 | 68.05 ± 0.93 | – | – |
| ResNet | 3.85 M | 70.07 ± 2.31 | 70.22 ± 1.84 | 51.71 ± 2.45 | 75.75 ± 0.93 |
| PatchTST | 616.58 K | 71.45 ± 1.79 | 70.83 ± 2.02 | 53.98 ± 2.83 | 68.81 ± 0.52 |
| **NeurOCNN** | 319.59 K | 75.22 ± 1.04 | 72.88 ± 1.37 | 56.37 ± 1.68 | 76.08 ± 0.64 |

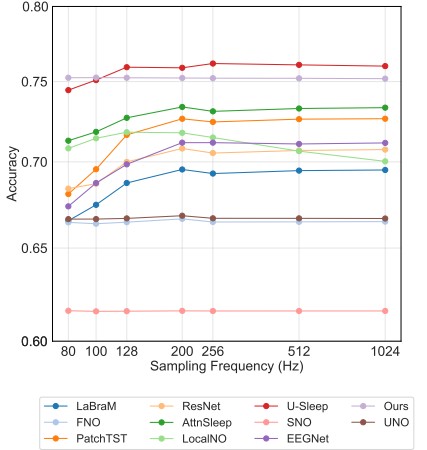

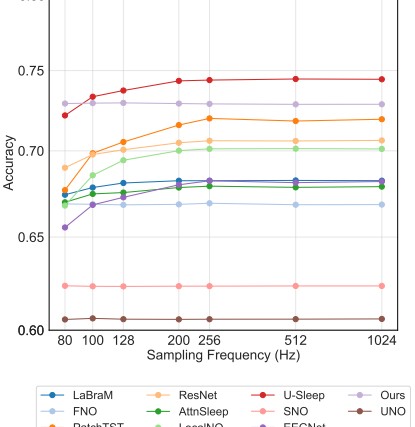

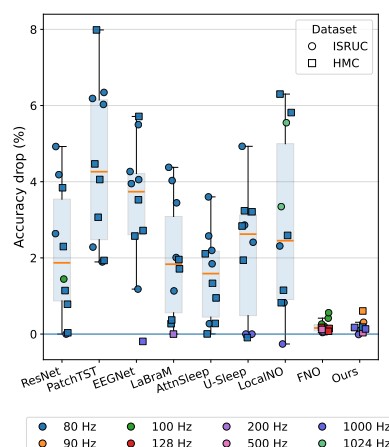

*(a)* Mean test accuracy (5 folds) versus test sampling rate on ISRUC.

*(b)* Mean test accuracy (5 folds) versus test sampling rate on HMC.

*(c)* Worst-case accuracy drop relative to the dataset-specific training $f_s$, reported across 5 folds on ISRUC and HMC.

*Figure 2.* Model performance under test-time sampling-rate shifts.

addition, we observe a consistent monotonic trend across methods: performance typically degrades as the test sampling frequency decreases, and the worst-case condition is most often attained at 80 Hz (the lowest frequency in $\mathcal{F}$), which dominates the maximum-drop statistic for many folds and models.

### 4.3. Noise Robustness

To evaluate model robustness, after training on clean data, we systematically corrupt the test set only using four distinct perturbations at a signal-to-noise ratio (SNR) of 5 dB: (i) powerline interference at 60 Hz along with its first two harmonics, (ii) baseline drift simulated via sinusoidal components sampled between 0.05-0.5 Hz, (iii) band-limited electromyographic (EMG) noise spanning 20-60 Hz, and

(iv) additive white Gaussian noise. We report both the absolute test accuracy across multiple sampling frequencies and the relative performance degradation compared to the clean baseline at each corresponding resolution. The comprehensive, detailed results for all conditions are provided in Appendix Table 5.

NeurOCNN consistently demonstrates superior robustness, exhibiting near-zero degradation under powerline and EMG noise across most test resolutions, and significantly smaller performance drops under Gaussian noise than competing methods. Conversely, the non-operator baselines suffer sharp performance declines when subjected to EMG and Gaussian noise. These trends persist across all evaluated test sampling frequencies, confirming that the observed robustness rankings are inherently architectural and not artifactual

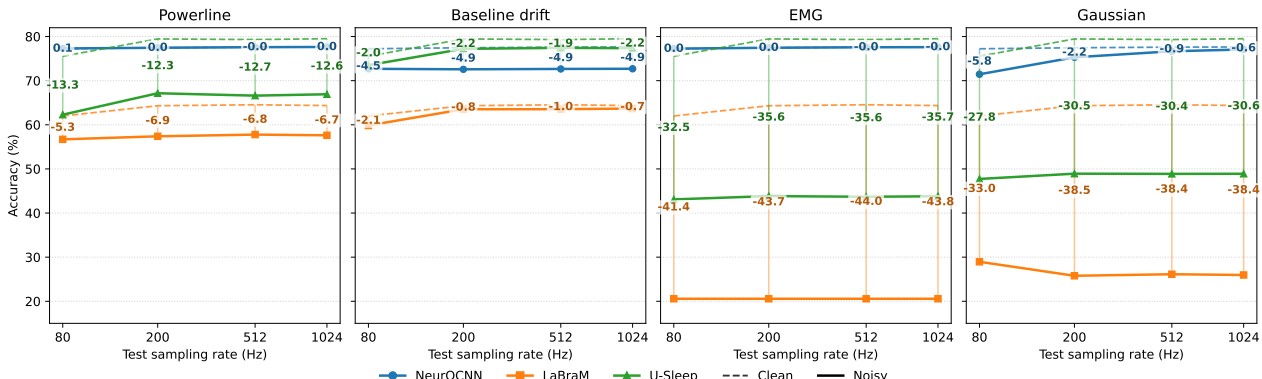

*Figure 3.* Robustness to test-time noise across sampling rates. Dashed lines indicate clean accuracy, solid lines indicate accuracy after adding noise, and vertical connectors show the clean-to-noisy change. Colored numeric labels report the change in accuracy in percentage points relative to the clean accuracy.

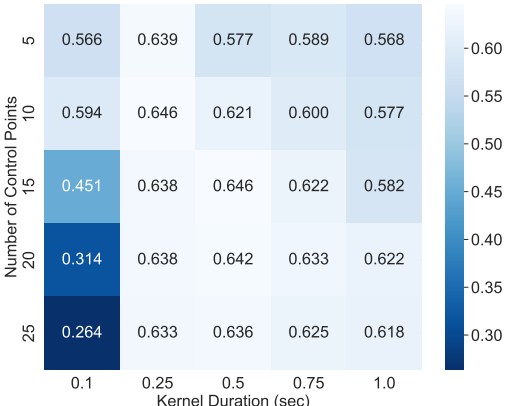

*Figure 4.* Ablation of SplineConv1d kernel duration and number of control points on the Sleep-EDF (EEG) dataset. The values indicate the mean accuracy.

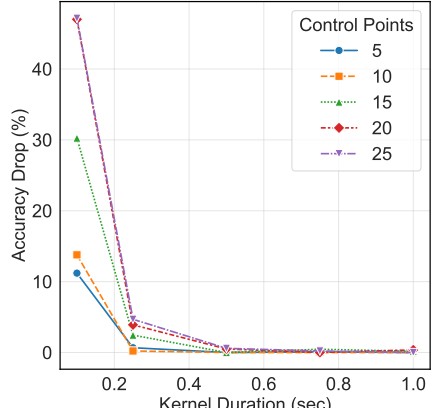

*Figure 5.* Effect of SplineConv1d kernel duration and number of control points on sampling-rate robustness on the Sleep-EDF (EEG) dataset.

properties of a specific discretization.

### 4.4. Ablation Studies

#### 4.4.1. SPLINE CONVOLUTION DESIGN

Figure 4 sweeps the spline-kernel convolutional operator over kernel duration and the number of spline control points (kernel degrees of freedom). Performance exhibits a clear optimal regime at moderate durations (0.25-0.75 s) with moderate control points (10-20), with peak performance near 0.5 s and 15 control points and a relatively flat high-performing neighborhood. Very short kernels (0.1 s) degrade sharply as control points increase, consistent with an expressivity-stability trade-off under discretization: increasing control points raises the effective bandwidth/curvature of the continuous kernel, but the kernel is ultimately sampled on the input grid, so overly flexible kernels on short support can induce sampling-rate sensitivity (aliasing-like distor-

tions on coarser grids) and overfit discretization-specific artifacts. Conversely, long kernels with too few control points become overly smooth and can underfit localized structure, explaining the weaker results away from the mid-duration, mid-capacity regime.

To complement the accuracy heatmap in Figure 4, we also quantify how kernel duration and number of control points impact sampling-rate robustness. Figure 5 shows that the worst-case accuracy drop is largest for short kernels and increases with number of control points at fixed kernel duration, while the mid-duration, mid-capacity regime remains comparatively stable across sampling frequencies. A controlled comparison between the proposed SplineConv operator and an MLP-parameterized continuous convolution baseline instantiated via CKConv (Romero et al., 2022) is reported in Appendix A.2.1.

*Table 2.* Fourier projection pooling ablation on ISRUC. $T_{seg}$ and $T_{epoch}$ are in seconds and $f_{max}$ is in Hz.

| $T_{\text{seg}}$ | $M$ | $f_{\max}$ | $T_{\text{epoch}}$ | Accuracy (%) |
|---|---|---|---|---|
| 5 | 100 | 9.9 | 153.5 | 76.03 |
| 15 | 300 | 9.967 | 156.3 | 74.46 |
| 30 | 600 | 9.98 | 185.2 | 75.89 |

#### 4.4.2. FOURIER PROJECTION POOLING DESIGN

We ablate the Fourier projection pooling design by varying the segment duration $T_{\text{seg}}$ and the number of basis terms $M$ while keeping the effective spectral coverage approximately fixed (i.e., matching $f_{\max}$ as defined in Sec. 3.4). Specifically, we choose three $(T_{\text{seg}}, M)$ configurations with nearly identical $f_{\max}$ (Table 2) and train NeurOCNN on ISRUC with all other hyperparameters unchanged.

The results show no meaningful difference in accuracy across these configurations when $f_{\max}$ is matched. In particular, $T_{\text{seg}}{=}5$s with $M{=}100$ achieves comparable accuracy to $T_{\text{seg}}{=}30$s with $M{=}600$ (76.03 vs. 75.89), while reducing training time per epoch ($T_{\text{epoch}}{=}153.5$s vs. 185.2s). These findings suggest that segmentation does not degrade performance under matched spectral coverage, and motivate using shorter segments as a computationally efficient default.

## 5. Conclusion

We introduced NeurOCNN, a neural-operator-inspired model for physiological time series that combines a continuous-time spline-kernel convolutional backbone with Fourier basis projection pooling and a lightweight classification head. Across multiple modalities and datasets, NeurOCNN achieves strong performance, matching or closely approaching specialized sleep-staging baselines and outperforming neural-operator backbones under controlled comparisons. This demonstrates that the proposed operator instantiation is well suited for capturing localized, nonstationary structure in physiological signals. A key empirical result is discretization robustness: when the test set is resampled across a wide range of sampling frequencies, NeurOCNN maintains comparatively stable accuracy with small worst-case drops relative to the training frequency. Ablations further support the design choices, highlighting a stable operating regime for the spline-kernel parameterization and showing that segmented Fourier projection pooling can be used as an efficient default without a measurable loss in accuracy when spectral coverage is matched.

Beyond physiological decoding, NeurOCNN's discretization robustness and ability to capture localized temporal structure suggest broader applicability to other continuous, nonstationary sensor signals. The method can be naturally extended to any domain governed by continuous, nonstationary physical processes observed via varying high-frequency sensor arrays. For instance, in engineering and physics, NeurOCNN could be applied to acoustic emission signals for structural health monitoring, seismic waveform characterization in geophysics, or high-frequency telemetry data in industrial Internet of Things (IoT) frameworks. By grounding localized convolutions in physical time rather than a fixed grid, the architecture provides a foundational paradigm for deploying robust, cross-resolution classifiers across diverse real-world time-series tasks.

**Limitations and Future Work.** While NeurOCNN is robust to noise and sampling-rate shifts, it cannot natively handle missing modalities or irregularly sampled data where $\Delta t_n$ varies non-uniformly. Because its backbone relies on continuous convolutions mapped to uniform observational grids, sporadic data dropouts require external alignment or interpolation prior to inference. Furthermore, our evaluation is limited to standardized cohorts, leaving resilience against multi-site clinical sensor variability unexamined.

Future work will focus on embedding adaptive, continuous-time interpolation layers directly into the spline framework to natively ingest non-uniform inputs, while validating the model on heterogeneous, multi-center clinical datasets and expanding its utility to regression tasks.

## Impact Statement

This work aims to advance machine learning methods for real-world physiological time series by using a neural-operator-based formulation to improve robustness under changes in signal discretization. More reliable biosignal modeling could support clinical monitoring and health research by reducing dependence on device-specific sampling rates and preprocessing pipelines. In practice, such models may help improve reproducibility across acquisition settings, lower the burden of adapting models to new devices, and enable more robust deployment across heterogeneous physiological settings.

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

# A. Appendix

## A.1. Dataset Details and Preprocessing

### A.1.1. OVERVIEW

We evaluate NeurOCNN on sleep staging across multiple modalities (EEG/EOG) and on rhythm classification from single-lead ECG. For sleep staging datasets, we represent each recording as a sequence of 30 s non-overlapping epochs with a unified 5-class label space $\{W, N1, N2, N3, R\}$. When datasets provide finer-grained stages (e.g., $\{3, 4\}$ in Rechtschaffen & Kales), we merge them into N3 and drop non-sleep and undefined annotations (e.g., movement, unknown) following standard practice. For ECG, we follow the PhysioNet/CinC 2017 label set $\{N, A, O, \sim\}$ and segment signals into fixed-length frames.

*Table 3.* Summary of datasets and the signal configurations used in our experiments. Sleep staging labels are mapped to a unified 5-class space (W/N1/N2/N3/R).

| Dataset | Task | Modality | Channels used | $f_s$ | Segment | #classes |
|---|---|---|---|---|---|---|
| Sleep-EDF (Expanded) | Sleep staging | EEG | Fpz–Cz, Pz–Oz | 100 Hz | 30 s | 5 |
| SleepEOG (derived from Sleep-EDF) | Sleep staging | EOG | EOG-horizontal | 100 Hz | 30 s | 5 |
| ISRUC (Subgroup I) | Sleep staging | EEG | C3–A2, C4–A1 | 200 Hz | 30 s | 5 |
| HMC Sleep Staging | Sleep staging | EEG | C4–M1, C3–M2 | 256 Hz | 30 s | 5 |
| PhysioNet/CinC 2017 ECG | Rhythm cls. | ECG | single lead | 300 Hz | 8 s | 4 |

### A.1.2. SLEEP-EDF (EEG) AND DERIVED SLEEPEOG

**Dataset.** Sleep-EDF Expanded is a large collection of whole-night PSG recordings released via PhysioNet, containing EEG (Fpz–Cz, Pz–Oz) and horizontal EOG channels with expert hypnograms (Kemp et al., 2000; Goldberger et al., 2000). We use (i) an EEG-only variant for sleep staging and (ii) an EOG-only benchmark (SleepEOG) derived from the same recordings. The EOG-only benchmark was used for model comparison, whereas the EEG-only benchmark was used for ablation studies.

**Preprocessing.** For each recording, we extract only the channels required for the target benchmark (EEG: Fpz–Cz and Pz–Oz; EOG: horizontal EOG) and align signals with the provided hypnogram annotations. For SleepEOG, we apply an EOG-specific band-pass filter from 0.3–35 Hz, whereas for EEG we keep filtering minimal and rely on *robust normalization* (median-centering and IQR-scaling per channel, with optional clipping to reduce the influence of extreme artifacts), consistent with the preprocessing pipeline adopted in (Perslev et al., 2021). Finally, we segment each recording into non-overlapping 30 s epochs and map stage labels to a unified 5-class space by merging deep sleep stages (S3, S4 $\rightarrow$ N3) and discarding movement and unknown epochs.

### A.1.3. ISRUC SLEEP (EEG)

**Dataset.** ISRUC-Sleep is a public clinical PSG dataset containing recordings from healthy subjects and subjects with sleep disorders, with sleep-stage labels (Khalighi et al., 2016). We use ISRUC Subgroup I in an EEG-only configuration.

**Preprocessing.** We extract two EEG derivations corresponding to the central leads (C3 and C4) referenced to mastoids or auricular electrodes. In ISRUC, the reference naming is not fully uniform across recordings (e.g., A1/A2 vs. M1/M2), and in a small number of cases the constituent channels are provided separately rather than as pre-computed referential derivations. To ensure a consistent representation, we standardize channel construction by (i) using the available referential derivations when present and otherwise (ii) computing the corresponding referenced signals from the constituent channels, while keeping the final channel ordering fixed. We then apply a low-pass filter at 35 Hz and perform per-channel robust normalization using median/IQR scaling. Recordings are segmented into contiguous 30 s epochs and ISRUC's 6-stage labels are mapped to the unified 5-class space by merging N4 into N3. To mitigate severe artifacts, we remove epochs whose maximum absolute amplitude exceeds a fixed threshold.

### A.1.4. HMC SLEEP STAGING (EEG)

**Dataset.** The Haaglanden Medisch Centrum (HMC) sleep staging database is a clinical PSG dataset released on PhysioNet with expert sleep-stage scoring (Alvarez-Estevez & Rijsman, 2022; 2021; Goldberger et al., 2000). We use an EEG-only

configuration.

**Preprocessing.** We extract two EEG channels (C4–M1 and C3–M2). If referential derivations are not directly available, we derive them from constituent references when possible. We apply a low-pass filter at 35 Hz and use per-channel robust normalization via median/IQR scaling. We segment recordings into non-overlapping 30 s epochs and align each epoch with the corresponding sleep-stage annotation from the scoring file (already in the 5-class scoring - {W, N1, N2, N3, R}). As with ISRUC, we exclude epochs with extreme amplitudes using a fixed max-absolute threshold.

### A.1.5. PHYSIONET/CINC 2017 ECG

**Dataset.** The PhysioNet/Computing in Cardiology Challenge 2017 dataset provides short single-lead ECG recordings labeled as normal rhythm (N), atrial fibrillation (A), other rhythm (O), or noisy (∼), sampled at 300 Hz (Clifford et al., 2017).

**Preprocessing.** We follow the framing protocol used by bioFAME (Liu et al., 2024). Specifically, we load the single-lead waveform from the provided training files and labels from the corresponding `REFERENCE.csv`. We then segment each recording into non-overlapping fixed-length frames (8 s at 300 Hz), dropping any final partial frame. In line with (Liu et al., 2024), we keep signal processing minimal by default. Each extracted frame inherits the recording-level label in {N, A, O, ∼}.

## A.2. Results

### A.2.1. SPLINECONV1D VS. MLP-BASED CONTINUOUS CONVOLUTION (CKCONV)

Figure 6 compares the proposed SplineConv operator (using the best-performing kernel duration/control-point setting from the previous ablation) against an *MLP-parameterized* continuous convolution baseline instantiated via CKConv (Romero et al., 2022). CKConv represents each kernel $k_{o,i}(\tau)$ implicitly through a small neural network evaluated at continuous offsets, i.e., $k_{o,i}(\tau) = \mathrm{MLP}_{o,i}(\tau)$, which is then discretized on the observed grid to perform convolution. We conduct a controlled architecture match by keeping the backbone depth, kernel duration, pooling, head, and all training settings fixed, and varying only the hidden-channel width of the CKConv kernel network (4/8/16/32). Across all test sampling frequencies, SplineConv achieves consistently higher accuracy and exhibits a flatter accuracy–frequency profile, indicating stronger discretization robustness. In contrast, the MLP-based CKConv remains below SplineConv for every width setting, and its performance becomes increasingly sampling-rate dependent as width increases: larger CKConv variants improve peak accuracy at favorable sampling rates, but incur larger drops at mismatched rates, widening the gap between best- and worst-case performance.

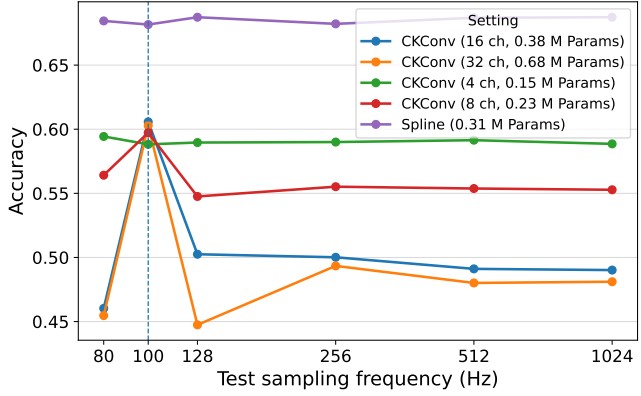

*Figure 6.* Controlled comparison between SplineConv1d and CKConv on the Sleep-EDF (EEG) dataset across test sampling rates. The backbone depth, kernel duration, pooling module, prediction head, and training settings are kept fixed, while the continuous-kernel parameterization is varied. SplineConv1d achieves higher accuracy and a flatter accuracy–frequency profile, indicating stronger sampling-rate robustness.

These results support our choice of spline parameterization over an unconstrained MLP kernel. Physiological time series are typically dominated by structured low-to-mid frequency content and localized temporal phenomena; after standard

preprocessing, the relevant discriminative patterns rarely require arbitrarily sharp or rapidly oscillatory filters. SplineConv enforces a compact-support, smooth ($C^2$) kernel class with a small number of interpretable degrees of freedom (control points over a fixed physical duration). This smoothness introduces an implicit spectral regularization: kernels with limited curvature cannot concentrate substantial energy at very high frequencies, yielding discrete realizations that remain stable when the sampling grid changes. In contrast, an MLP kernel parameterization is substantially more flexible and can realize sharply varying or oscillatory kernels within the same support; as the hidden width increases, CKConv gains capacity to fit discretization-specific structure that may be effective on the native training grid but is not preserved under grid changes. When evaluated at a different sampling rate, the discretized kernel and its interaction with the sampled signal can be attenuated or distorted, producing larger accuracy degradation. This capacity–inductive-bias trade-off explains (i) the lower overall performance of CKConv under identical backbone/pooling/head choices, and (ii) why widening the MLP kernel (16–32 channels) can reduce sampling-rate robustness even as parameter count increases.

### A.2.2. TEST PERFORMANCE ACROSS SAMPLING FREQUENCIES

Table 4 reports the test accuracy (mean $\pm$ standard deviation over 5 folds) for each dataset. The colored column marks the dataset's training sampling frequency (i.e., the discretization used during training), and the remaining columns quantify generalization under sampling-rate shift at test time. Across ISRUC, HMC, Sleep-EDF (EOG), and ECG, we observe that performance is strongly model-dependent under sampling-rate shift: neural-operator baselines (FNO, U-NO, SNO, LocalNO) tend to cluster at lower accuracies and remain relatively stable across frequencies but underperforming, whereas stronger sequence architectures and modality-specific baselines (U-Sleep, AttnSleep, ResNet, PatchTST) achieve higher accuracy but can exhibit varying sensitivity when evaluated at different sampling rates. Overall, NeurOCNN remains consistently competitive at all evaluated frequencies while remaining robust to changes in the sampling rate at test time.

### A.2.3. ROBUSTNESS TO TEST-TIME NOISE

Table 5 evaluates the robustness of NeurOCNN and baseline models under four types of test-time noise across multiple sampling resolutions. For each setting, the table reports the noisy test accuracy together with the accuracy drop relative to clean performance at the same resolution, allowing robustness to be compared independently of differences in clean accuracy.

*Table 4.* **Test performance across sampling frequencies (Hz).** Values are mean test accuracy (%) $\pm$ standard deviation (in percentage points, pp) over 5-fold cross-validation. Fourier Pooling is denoted by FP. The shaded column corresponds to the training sampling rate. Best accuracy and second-best accuracy are color-coded.

| | | | | ISRUC | | | |
|---|---|---|---|---|---|---|---|
| **Model** | **200** | **80** | **100** | **128** | **256** | **512** | **1024** |
| FNO + FP | $66.65 \pm 2.03$ | $66.45 \pm 2.10$ | $66.37 \pm 2.15$ | $66.46 \pm 2.09$ | $66.47 \pm 2.07$ | $66.48 \pm 2.06$ | $66.50 \pm 2.06$ |
| U-NO + FP | $66.83 \pm 1.27$ | $66.64 \pm 1.35$ | $66.64 \pm 1.36$ | $66.68 \pm 1.30$ | $66.68 \pm 1.33$ | $66.68 \pm 1.33$ | $66.70 \pm 1.34$ |
| SNO + FP | $61.58 \pm 2.72$ | $61.59 \pm 2.80$ | $61.56 \pm 2.77$ | $61.57 \pm 2.74$ | $61.58 \pm 2.73$ | $61.58 \pm 2.73$ | $61.58 \pm 2.74$ |
| LocalNO + FP | $71.77 \pm 2.33$ | $70.82 \pm 1.67$ | $71.43 \pm 1.44$ | $71.81 \pm 1.72$ | $71.48 \pm 2.08$ | $70.65 \pm 2.38$ | $70.03 \pm 2.81$ |
| EEGNet | $71.16 \pm 3.02$ | $67.37 \pm 3.35$ | $68.74 \pm 3.30$ | $69.85 \pm 3.09$ | $71.16 \pm 3.02$ | $71.08 \pm 3.01$ | $71.15 \pm 3.01$ |
| AttnSleep | $73.38 \pm 1.78$ | $71.28 \pm 1.94$ | $71.83 \pm 1.94$ | $72.70 \pm 1.88$ | $73.10 \pm 1.89$ | $73.28 \pm 1.84$ | $73.34 \pm 1.84$ |
| U-Sleep | $75.89 \pm 1.05$ | $74.45 \pm 2.86$ | $75.10 \pm 2.47$ | $75.93 \pm 1.76$ | $76.17 \pm 1.31$ | $76.08 \pm 1.15$ | $76.00 \pm 1.09$ |
| LaBraM | $69.54 \pm 1.65$ | $66.54 \pm 1.82$ | $67.46 \pm 1.76$ | $68.74 \pm 1.70$ | $69.30 \pm 1.65$ | $69.47 \pm 1.64$ | $69.51 \pm 1.64$ |
| PatchTST | $72.64 \pm 1.56$ | $68.09 \pm 2.63$ | $69.56 \pm 2.41$ | $71.64 \pm 1.93$ | $72.44 \pm 1.69$ | $72.62 \pm 1.67$ | $72.64 \pm 1.60$ |
| ResNet | $70.81 \pm 1.96$ | $68.41 \pm 3.36$ | $68.70 \pm 3.01$ | $70.00 \pm 2.51$ | $70.53 \pm 2.12$ | $70.69 \pm 2.06$ | $70.74 \pm 1.98$ |
| **NeurOCNN** | $75.23 \pm 1.04$ | $75.25 \pm 0.99$ | $75.25 \pm 1.00$ | $75.25 \pm 1.00$ | $75.22 \pm 1.07$ | $75.21 \pm 1.03$ | $75.19 \pm 1.04$ |

| | | | | HMC | | | |
|---|---|---|---|---|---|---|---|
| **Model** | **256** | **80** | **100** | **128** | **200** | **512** | **1024** |
| FNO + FP | $66.92 \pm 1.22$ | $66.88 \pm 1.20$ | $66.86 \pm 1.22$ | $66.83 \pm 1.19$ | $66.85 \pm 1.20$ | $66.83 \pm 1.21$ | $66.84 \pm 1.22$ |
| U-NO + FP | $60.57 \pm 7.54$ | $60.56 \pm 7.44$ | $60.61 \pm 7.50$ | $60.57 \pm 7.48$ | $60.56 \pm 7.50$ | $60.57 \pm 7.49$ | $60.58 \pm 7.44$ |
| SNO + FP | $62.32 \pm 1.67$ | $62.33 \pm 1.70$ | $62.31 \pm 1.68$ | $62.30 \pm 1.69$ | $62.31 \pm 1.70$ | $62.33 \pm 1.66$ | $62.33 \pm 1.67$ |
| LocalNO + FP | $70.12 \pm 1.67$ | $66.78 \pm 2.70$ | $68.54 \pm 2.11$ | $69.44 \pm 1.80$ | $70.01 \pm 1.71$ | $70.13 \pm 1.64$ | $70.11 \pm 1.65$ |
| EEGNet | $68.22 \pm 5.01$ | $65.54 \pm 3.15$ | $66.82 \pm 3.64$ | $67.26 \pm 3.89$ | $67.99 \pm 4.47$ | $68.11 \pm 4.69$ | $68.18 \pm 4.85$ |
| AttnSleep | $67.91 \pm 1.58$ | $66.97 \pm 1.55$ | $67.46 \pm 1.52$ | $67.53 \pm 1.54$ | $67.83 \pm 1.60$ | $67.84 \pm 1.57$ | $67.88 \pm 1.60$ |
| U-Sleep | $74.39 \pm 2.38$ | $72.16 \pm 2.53$ | $73.33 \pm 2.36$ | $73.72 \pm 2.34$ | $74.34 \pm 2.32$ | $74.46 \pm 2.37$ | $74.44 \pm 2.38$ |
| LaBraM | $68.22 \pm 1.07$ | $67.41 \pm 0.91$ | $67.83 \pm 0.83$ | $68.09 \pm 0.84$ | $68.22 \pm 1.07$ | $68.24 \pm 1.07$ | $68.23 \pm 1.06$ |
| PatchTST | $71.98 \pm 1.75$ | $67.68 \pm 2.81$ | $69.85 \pm 2.40$ | $70.54 \pm 2.23$ | $71.57 \pm 1.92$ | $71.82 \pm 1.84$ | $71.93 \pm 1.84$ |
| ResNet | $70.61 \pm 1.88$ | $68.99 \pm 2.10$ | $69.78 \pm 1.91$ | $70.07 \pm 1.87$ | $70.49 \pm 1.87$ | $70.60 \pm 1.82$ | $70.63 \pm 1.81$ |
| **NeurOCNN** | $72.88 \pm 1.38$ | $72.90 \pm 1.47$ | $72.93 \pm 1.45$ | $72.94 \pm 1.44$ | $72.90 \pm 1.37$ | $72.85 \pm 1.31$ | $72.86 \pm 1.29$ |

| | | | | Sleep-EDF (EOG) | | | |
|---|---|---|---|---|---|---|---|
| **Model** | **100** | **80** | **128** | **200** | **256** | **512** | **1024** |
| FNO + FP | $48.91 \pm 0.97$ | $48.62 \pm 0.70$ | $48.57 \pm 0.85$ | $48.49 \pm 0.80$ | $48.57 \pm 0.85$ | $48.59 \pm 0.85$ | $48.56 \pm 0.83$ |
| U-NO + FP | $47.15 \pm 2.15$ | $46.82 \pm 2.28$ | $46.82 \pm 2.24$ | $46.80 \pm 2.29$ | $46.83 \pm 2.32$ | $46.84 \pm 2.36$ | $46.86 \pm 2.41$ |
| SNO + FP | $40.91 \pm 1.20$ | $40.90 \pm 1.11$ | $41.01 \pm 1.11$ | $41.00 \pm 1.11$ | $41.00 \pm 1.12$ | $41.05 \pm 1.08$ | $41.07 \pm 1.11$ |
| LocalNO + FP | $56.53 \pm 1.49$ | $54.32 \pm 2.74$ | $55.49 \pm 2.20$ | $54.78 \pm 2.38$ | $55.07 \pm 2.03$ | $54.54 \pm 1.97$ | $54.40 \pm 1.86$ |
| AttnSleep | $57.33 \pm 1.33$ | $55.22 \pm 1.74$ | $56.71 \pm 1.39$ | $56.71 \pm 1.41$ | $57.08 \pm 1.33$ | $57.22 \pm 1.36$ | $57.29 \pm 1.34$ |
| U-Sleep | $51.65 \pm 1.94$ | $50.39 \pm 2.50$ | $51.22 \pm 2.36$ | $51.33 \pm 2.23$ | $51.55 \pm 2.08$ | $51.73 \pm 2.04$ | $51.71 \pm 2.02$ |
| PatchTST | $54.12 \pm 3.60$ | $53.22 \pm 1.77$ | $53.92 \pm 2.34$ | $53.93 \pm 2.40$ | $54.32 \pm 3.12$ | $54.22 \pm 3.20$ | $54.19 \pm 3.36$ |
| ResNet | $52.11 \pm 2.11$ | $50.45 \pm 2.99$ | $51.81 \pm 2.74$ | $51.61 \pm 2.65$ | $51.92 \pm 2.48$ | $52.00 \pm 2.33$ | $52.04 \pm 2.28$ |
| **NeurOCNN** | $56.67 \pm 1.51$ | $56.73 \pm 1.78$ | $56.60 \pm 1.79$ | $56.38 \pm 1.77$ | $56.23 \pm 1.72$ | $56.14 \pm 1.76$ | $56.03 \pm 1.79$ |

| | | | | ECG | | | |
|---|---|---|---|---|---|---|---|
| **Model** | **300** | **80** | **100** | **128** | **256** | **512** | **1024** |
| FNO + FP | $60.33 \pm 4.29$ | $60.26 \pm 4.15$ | $60.26 \pm 4.16$ | $60.31 \pm 4.25$ | $60.31 \pm 4.25$ | $60.32 \pm 4.28$ | $60.32 \pm 4.28$ |
| U-NO + FP | $59.42 \pm 2.56$ | $59.41 \pm 2.54$ | $59.41 \pm 2.54$ | $59.43 \pm 2.59$ | $59.43 \pm 2.59$ | $59.42 \pm 2.57$ | $59.41 \pm 2.54$ |
| SNO + FP | $58.16 \pm 0.07$ | $58.16 \pm 0.07$ | $58.16 \pm 0.07$ | $58.16 \pm 0.07$ | $58.16 \pm 0.07$ | $58.16 \pm 0.07$ | $58.16 \pm 0.07$ |
| LocalNO + FP | $65.96 \pm 1.20$ | $65.56 \pm 1.01$ | $65.76 \pm 0.96$ | $65.81 \pm 0.99$ | $65.93 \pm 1.19$ | $65.84 \pm 1.20$ | $65.83 \pm 1.34$ |
| PatchTST | $68.90 \pm 0.53$ | $68.67 \pm 0.64$ | $68.80 \pm 0.49$ | $68.82 \pm 0.51$ | $68.84 \pm 0.51$ | $68.85 \pm 0.55$ | $68.86 \pm 0.53$ |
| ResNet | $75.87 \pm 0.96$ | $75.39 \pm 0.87$ | $75.70 \pm 0.88$ | $75.77 \pm 0.91$ | $75.79 \pm 1.00$ | $75.84 \pm 0.96$ | $75.87 \pm 0.96$ |
| **NeurOCNN** | $76.39 \pm 0.65$ | $75.34 \pm 0.69$ | $75.79 \pm 0.63$ | $76.00 \pm 0.73$ | $76.32 \pm 0.72$ | $76.42 \pm 0.62$ | $76.47 \pm 0.55$ |

*Table 5.* Robustness to test-time noise across resolutions. We report accuracy (%) and the change relative to clean accuracy at the same test resolution in parentheses. Smaller accuracy drops indicate stronger robustness. Lowest drop and second-lowest drop are color-coded.

| Test fs | Model | Clean | PL | BL | EMG | Gaus. |
|---|---|---|---|---|---|---|
| 80 | **NeurOCNN** | 77.23 | 77.30 (+0.07) | 72.69 (-4.55) | 77.26 (+0.02) | 71.45 (-5.78) |
| | EEGNet | 67.99 | 60.53 (-7.45) | 65.90 (-2.09) | 23.29 (-44.70) | 42.21 (-25.77) |
| | AttnSleep | 75.53 | 74.32 (-1.21) | 69.43 (-6.10) | 50.27 (-25.26) | 69.12 (-6.41) |
| | LaBraM | 62.00 | 56.70 (-5.31) | 59.85 (-2.15) | 20.57 (-41.43) | 28.95 (-33.06) |
| | U-Sleep | 75.54 | 62.29 (-13.25) | 73.52 (-2.02) | 43.09 (-32.45) | 47.73 (-27.81) |
| 100 | **NeurOCNN** | 77.46 | 67.62 (-9.84) | 72.71 (-4.75) | 77.43 (-0.03) | 72.84 (-4.63) |
| | EEGNet | 68.93 | 60.63 (-8.29) | 66.94 (-1.99) | 23.18 (-45.11) | 41.78 (-27.14) |
| | AttnSleep | 75.70 | 74.29 (-1.41) | 69.69 (-6.00) | 49.65 (-26.05) | 68.80 (-6.90) |
| | LaBraM | 62.71 | 57.29 (-5.43) | 60.85 (-1.86) | 20.57 (-42.14) | 28.38 (-34.33) |
| | U-Sleep | 76.55 | 63.62 (-12.93) | 74.51 (-2.04) | 43.32 (-33.23) | 48.07 (-28.48) |
| 128 | **NeurOCNN** | 77.51 | 77.51 (0.00) | 72.66 (-4.85) | 77.51 (0.00) | 73.93 (-3.57) |
| | EEGNet | 70.68 | 60.85 (-9.83) | 68.65 (-2.03) | 22.90 (-47.79) | 40.81 (-29.87) |
| | AttnSleep | 75.77 | 74.38 (-1.39) | 70.54 (-5.22) | 48.64 (-27.11) | 67.99 (-7.76) |
| | LaBraM | 63.95 | 57.92 (-6.03) | 62.08 (-1.87) | 20.57 (-43.37) | 27.14 (-36.82) |
| | U-Sleep | 78.31 | 64.96 (-13.35) | 76.34 (-1.98) | 43.62 (-34.69) | 48.65 (-29.67) |
| 200 | **NeurOCNN** | 77.46 | 77.46 (0.00) | 72.58 (-4.88) | 77.46 (0.00) | 75.30 (-2.16) |
| | EEGNet | 72.31 | 59.84 (-12.47) | 69.88 (-2.43) | 22.66 (-49.65) | 39.05 (-33.26) |
| | AttnSleep | 75.76 | 73.93 (-1.84) | 70.74 (-5.02) | 47.29 (-28.47) | 66.87 (-8.89) |
| | LaBraM | 64.32 | 57.39 (-6.93) | 63.52 (-0.80) | 20.57 (-43.74) | 25.78 (-38.54) |
| | U-Sleep | 79.46 | 67.15 (-12.31) | 77.21 (-2.25) | 43.84 (-35.62) | 48.92 (-30.53) |
| 256 | **NeurOCNN** | 77.63 | 77.63 (0.00) | 72.71 (-4.91) | 77.64 (+0.01) | 76.06 (-1.56) |
| | EEGNet | 71.62 | 60.17 (-11.45) | 69.36 (-2.26) | 22.76 (-48.87) | 39.87 (-31.75) |
| | AttnSleep | 75.82 | 74.24 (-1.58) | 70.56 (-5.26) | 47.87 (-27.95) | 67.37 (-8.45) |
| | LaBraM | 64.42 | 57.95 (-6.47) | 63.19 (-1.23) | 20.57 (-43.84) | 26.41 (-38.01) |
| | U-Sleep | 79.11 | 66.12 (-12.99) | 77.32 (-1.79) | 43.73 (-35.38) | 48.81 (-30.30) |
| 512 | **NeurOCNN** | 77.60 | 77.59 (-0.01) | 72.66 (-4.94) | 77.57 (-0.03) | 76.66 (-0.94) |
| | EEGNet | 72.09 | 60.04 (-12.05) | 69.72 (-2.37) | 22.73 (-49.36) | 39.50 (-32.59) |
| | AttnSleep | 75.77 | 74.13 (-1.64) | 70.83 (-4.94) | 47.64 (-28.13) | 67.13 (-8.64) |
| | LaBraM | 64.54 | 57.78 (-6.75) | 63.54 (-1.00) | 20.57 (-43.96) | 26.13 (-38.41) |
| | U-Sleep | 79.32 | 66.61 (-12.71) | 77.44 (-1.88) | 43.69 (-35.63) | 48.88 (-30.44) |
| 1024 | **NeurOCNN** | 77.64 | 77.64 (0.00) | 72.70 (-4.94) | 77.60 (-0.04) | 77.08 (-0.56) |
| | EEGNet | 72.21 | 59.99 (-12.22) | 69.94 (-2.27) | 22.71 (-49.50) | 39.24 (-32.97) |
| | AttnSleep | 75.80 | 74.01 (-1.79) | 70.81 (-4.99) | 47.42 (-28.37) | 66.96 (-8.85) |
| | LaBraM | 64.36 | 57.62 (-6.74) | 63.68 (-0.68) | 20.57 (-43.79) | 25.96 (-38.40) |
| | U-Sleep | 79.51 | 66.92 (-12.58) | 77.36 (-2.14) | 43.80 (-35.71) | 48.90 (-30.61) |

PL: powerline (5 dB), BL: baseline drift (5 dB), EMG: electromyographic noise (5 dB), Gaus.: white Gaussian noise (5 dB).

