# OpenReview forum: "NeurOCNN: A Neural-Operator-Based Model for Physiological Time Series"
_ICML.cc/2026/Conference — ICML 2026 regular_

### Official Review · Reviewer_GnLi · 2026-03-02

**Soundness:** 3
**Presentation:** 3
**Significance:** 3
**Originality:** 2
**Overall Recommendation:** 5
**Confidence:** 4

**Summary:**

This paper proposes a series of architectural innovations for modelling physiological signals. Their main argument underlying these innovations is that physiological signals should be viewed and treated like a continuous signal. The different elements of their modal are thouroughly evaluated on an extensive benchmark.

**Compliance With Llm Reviewing Policy:**

Affirmed.

**Key Questions For Authors:**

See above.

**Limitations:**

See above.

**Strengths And Weaknesses:**

The paper's important contributions lie in its architectural novelties: the authors introduce a novel operator based architecture for learning from physiological signals. They carry out an extensive benchmark against concurrent models, including domain specific ones, and show that their model outperforms most models, except for USeelp whose performance it closely matches (but with a order of magnitude less parameters!). An interesting property of their model is its strong robustness against irregular low-frequency sampling. I am however less convinced by their depiction of this property as being emergent, since they do not analyze when and how it appears (i.e. is it conditionned on using a sufficient number of samples, of training iterations, ...).

Overall, the paper is well written and does not overclaim its contributions. The proposed model seems to reach excellent performance against a wide variety of methods, making it a solid proposal for the considered task. Notably, its limited number of parameters is a strong selling point. The ablations motivate all architectural innovations introduced in the paper.

As the authors admit in their conclusion, a blind spot of the paper is the evaluation of their model under distribution shifts, which are ubiquitous in sleep staging. I would strongly recommend adding a series of experiments to analyze how well the model does perform here, and if its performance gap against large foundation models such as LabraM is still present.

I would also appreciate a longer discussion of "sampling rate robustness as an emergent property": is this property present in the model from the early training stages i.e. is it solely a result of architectural choices? Does it emerge during training? Depending on the answers to these questions, I would suggest rephrasing this property as an inductive bias of the architecture, rather than an emergent property.

I believe that your problem statement is very similar to the one given in a series of articles by Bleistein et al. - see for instance "Learning the dynamics of sparsely observed interacting systems", "On the generalization and approximation capacities of neural controlled differential equations" and "dynamic survival analysis with controlled latent states". You could consider citing these articles in your introduction, as a) they drive the same point than you i.e. the observed signal is a discrete counterpart of the unobserved continuous process, and b) provide a statistical analysis of the induced bias.

---

> ### Author Rebuttal · Authors · 2026-03-31
>
> We sincerely thank the reviewer for their thoughtful and constructive feedback. We address the reviewer’s main points below.
>
> > **Model evaluation under distribution shifts**
>
> We agree that robustness under distribution shift is important for sleep staging. In the original submission, our primary focus was one practically important shift: test-time sampling-rate mismatch, where the model is trained at the native dataset sampling rate and evaluated zero-shot across unseen resolutions. This is itself a form of distribution shift, and NeurOCNN was specifically designed for this setting.
>
> Our sleep-staging experiments also use subject-level train/test separation, so the reported results already reflect generalization to unseen subjects, rather than same-subject window splitting.
>
> We also performed additional test-time noise robustness experiments on ISRUC under multiple unseen resolutions (80, 512, and 1024 Hz), without changing training. We refer the reviewer to our response to Reviewer fJsf, where the full table is provided. In that experiment, we evaluate powerline noise, baseline drift, EMG contamination, and white Gaussian noise added only at test time. These results show that NeurOCNN remains substantially more stable than LaBraM and U-Sleep under several common corruptions, with minimal change under powerline noise and EMG and only limited degradation under baseline drift and Gaussian noise. In this sense, the model shows robustness not only to sampling-rate shift, but also to a broader class of test-time distribution shifts arising from signal corruption.
>
> At the same time, we agree that broader shifts, especially cross-dataset, cross-device, and structured sensor variability, remain important directions for future work, and we will clarify this in the revised manuscript.
>
> > **Sampling rate robustness as an emergent property**
>
> Sampling-rate robustness in NeurOCNN follows from the architectural choices. The continuous-time spline convolution parameterizes kernels over physical time, and the Fourier projection pooling uses time-referenced integration, so the model’s computation is not tied to a fixed discretization. This gives the architecture a built-in bias toward consistent behaviour across sampling rates, even without any explicit invariance objective. Consistent with this interpretation, the model shows a nearly flat accuracy profile across test-time sampling rates even early in training. After 1 epoch on the ISRUC dataset, accuracy is already stable from 80 to 1024 Hz, with a maximum drop of only 0.05 percentage points relative to the training sampling frequency of 200 Hz. After 5 epochs, this same flat profile is preserved while overall accuracy improves, with a maximum drop of only 0.10 percentage points. All values in the table are reported in %, and the accuracy drop is computed as the difference between the accuracy at 200 Hz and the lowest accuracy across the evaluated test resolutions.
>
> | Epoch |    80 |   100 |   128 |   200 |   256 |   512 |  1024 | Acc. Drop |
> | ----- | ----: | ----: | ----: | ----: | ----: | ----: | ----: | --------: |
> | 1     | 73.57 | 73.64 | 73.71 | 73.62 | 73.66 | 73.65 | 73.73 |      **0.05** |
> | 5     | 76.73 | 76.74 | 76.76 | 76.67 | 76.63 | 76.60 | 76.57 |      **0.10** |
>
> We nevertheless understand that the phrase “emergent property” may cause confusion, and we will revise this wording in the updated manuscript.
>
> > **Connection to Bleistein et al. and Related Continuous-Time Modelling Work**
>
> Thank you for pointing us to these references. We agree that the articles from Bleistein et al. are relevant prior work for our problem formulation, particularly in viewing the observed sequence as a discrete counterpart of an underlying continuous process and in analyzing bias induced by sampling and discretization. We will cite these articles in our updated manuscript.

---

> > ### Author Rebuttal · Reviewer_GnLi · 2026-04-01
> >
> > That answers my questions, thanks. I still believe that the phrasing of "emerging property" is a slight overclaim. I have nevertheless increased my score.

---

> > > ### Author Response · Authors · 2026-04-05
> > >
> > > We thank the reviewer again for their valuable comments and suggestions. In the revised manuscript, we will revise the phrasing and refer to this property as an inductive bias of the architecture rather than an emergent property.

---

### Official Review · Reviewer_uB3G · 2026-03-06

**Soundness:** 4
**Presentation:** 3
**Significance:** 4
**Originality:** 3
**Overall Recommendation:** 5
**Confidence:** 3

**Summary:**

The authors introduce **NeurOCNN** as a neural-operator-based model for analysing physiological time series that generalises across datasets and sampling rates.

More specifically, a *function-to-label* operator is learned in continuous time, which enables evaluation on different discretisations. This operator is implemented via a spline convolution method that learns continuous filters and yields time-indexed features of varying size (depending on sampling rate).
A fixed-length representation is obtained via segment-wise Fourier projection, which is used by an attention-based task-head for downstream tasks.

The effectiveness of NeurOCNN is empirically validated in experiments against two groups of baselines across 5 datasets: the first group varies with other state-of-the-art neural operators, while the second group consists of state-of-the-art non-operator models widely used for time-series analysis.
NeurOCNN matches state-of-the-art accuracy of task-specific baselines, with lower variability and a smaller parameter footprint. Moreover, the convolutional operator employed yields significant gains over other neural operators.
Discretisation robustness is tested by reporting test accuracy as a function of sampling frequency, in which NeurOCNN exhibits less variation across frequencies, whereas other methods show decreases in accuracy as the sampling rate decreases.
Finally, ablation study highlights optimal hyperparameters for the spline convolution and Fourier projection modules.

**Compliance With Llm Reviewing Policy:**

Affirmed.

**Final Justification:**

My questions were answered, and there was little to address in the paper to begin with.

**Key Questions For Authors:**

1. In what ways does changing temporal resolution affect models? Could you be more specific (and perhaps give examples)? If the underlying phenomenon is continuous, then changing the sampling rate shouldn’t affect its presence (as long as it satisfies the Nyquist condition), right?
2. What’s the motivation behind opting for a convolutional operator? Additionally, could you elaborate on how spline convolutions function and what their benefits are (over naive convolutions)?
3. You introduce 5 datasets in section 3.6, but your results table only contains the results for 4: ISRUC, HMC, Sleep-EDF (EOG), and ECG. Is it just me or are the results for Sleep-EDF (EEG) missing?

**Limitations:**

Yes

**Strengths And Weaknesses:**

1. Soundness
The submission claims to introduce a neural-operator based model for physiological time-series analysis, NeurOCNN, as a more generalisable model that exhibits discretisation invariance.
Overall, these claims are well supported through empirical results, in which NeurOCNN outperforms specialised baselines in terms of accuracy, while remaining relatively lightweight. Moreover, these results are consistent across sampling frequencies, whereas baseline models show diminished performance with lower sampling rates, supporting the discretisation invariance. Only the generalisability claim could be better supported as the scope of datasets and tasks covered is relatively narrow, though this limitation is already identified by the authors.
The methodology itself appears to be sound; while at times hard to follow in the text, in part due to my limited knowledge of neural operators, the authors did attach a code repository which is quite helpful and strengthens the soundness of their work.
2. Presentation
Overall, the paper is well written and structured. Only the method section could be improved upon as it is rather dense. In part this is to be expected given the intricate modelling that is applied, and in part due to my limited knowledge of neural operators, yet I would argue that there is room to improve the writing for clarity, particularly sections 3.3 and 3.4.
For starters, method subsections could benefit from explicitly referring to parts in Figure 1 (which is currently left unaddressed in the text). Sections in the text are also over-notated: for instance, equation (14) is unnecessary, with $I_s$ not being used anywhere else. Subsections 3.6-3.8 should have their own “Experimental Design” section or something similar distinguishing it from the modelling approach.
Minor issues:
    - (Abstract) “while exhibiting discretization invariance.” is followed by “enabling robust inference under sampling-rate shifts.”, which is somewhat redundant. Could benefit from more explicitly stating what each part of NeurOCNN enables.
    - (Section 2) Two big laps of text can be hard to parse, would benefit from splitting up into paragraphs.
    - (Neural Operators Section) Left a random “5” in at the end.
    - (Section 3.6) The descriptions in the appendix are excellent, but here it would still be nice to see a concise overview of the datasets. As it stands currently, it is not directly obvious which datasets are used.
    - (Table 1) Could you **bold** the best results (and optionally *italics* for the 2nd best) for clarity?
    - (Section 4.2) Surely formally defining how to compute the mean in equation (20) is not needed.
    - (Conclusion) This section is also supposed to have a section number, I believe.
3. Significance
The paper addresses the problem of time-series modelling, which remains an ever-relevant challenge, especially for physiological data. To do so, neural operators are used, which represent a novel and promising direction in machine learning. By adapting this method for physiological data, which requires a function-to-label operator as opposed to typical function-to-function, it sets up future works to incorporate neural operators for discretisation invariance.
How broadly this approach may be built on remains somewhat to be seen as more research can be done on more varied data and tasks.
4. Originality
The paper is original, incorporating novel neural operators in a domain that had yet to benefit from them. The method mainly combines several existing techniques, providing good, though at times a bit obtuse, reasoning behind their choices.

---

> ### Author Rebuttal · Authors · 2026-03-31
>
> We sincerely thank the reviewer for their thoughtful and constructive feedback. We address the reviewer’s main points below.
>
> > **In what ways does changing temporal resolution affect models?**
>
> We agree that the underlying phenomenon is continuous and the signal is sufficiently band-limited, thus, the information remains invariant under the Nyquist condition. However, model performance is sensitive to the discrete representation of that signal. Standard architectures operate on fixed sample patterns rather than the continuous-time phenomenon. Even with ideal resampling, an interpolated signal need not match the discrete sample statistics of a signal natively acquired on the training grid, since some values are synthesized by interpolation/filtering rather than directly observed [1]. Because kernels, strides, and pooling operations are defined in sample-space rather than physical-time, small changes in sample alignment and local phase can significantly alter internal activations [2], leading to the accuracy drops observed in our experiments.
>
> > **What’s the motivation behind opting for a convolutional operator?**
>
> We chose a convolutional operator because discriminative patterns in physiological time series are often local in time, such as transient waveform morphology, short oscillations, and neighborhood-level temporal context. Convolution provides a natural inductive bias for this setting through locality and weight sharing, making it effective for extracting local temporal features. In contrast, more global operator parameterizations such as Fourier Neural Operator emphasize global structure and may be less well matched to tasks where prediction depends primarily on local waveform patterns.
>
> The spline convolution implements a continuous-time kernel parameterization rather than learning a separate discrete kernel for each sampling rate. Specifically, for each input-output channel pair, the model learns a small set of spline control points over a normalized temporal support. At inference time, given the sampling frequency $f_s$, the kernel length is computed from a fixed kernel duration in seconds, and the continuous kernel is sampled on the corresponding discrete grid using natural cubic spline interpolation. The resulting sampled kernel is then applied with standard 1D convolution. Because the convolution approximates a continuous-time integral, we multiply the output by $dt = 1/f_s$. In this way, the kernel is defined in physical time rather than in a fixed number of samples.
>
> Compared with a naive discrete convolution, the spline parameterization offers two key advantages. First, it defines the kernel in continuous time, so the same learned operator can be evaluated consistently across sampling rates by sampling the kernel on the appropriate grid. Second, it acts as a useful regularizer: representing the kernel by a limited number of spline control points over a fixed temporal support constrains its effective bandwidth and prevents overly oscillatory filters, as also supported by our ablation study.
>
> > **Are the results for Sleep-EDF (EEG) missing?**
>
> The reviewer is correct that Sleep-EDF (EEG) does not appear in the main performance comparison table. This was intentional rather than an omission. Sleep-EDF (EEG), ISRUC, and HMC are structurally very similar benchmarks: all three are two-channel EEG sleep-staging tasks with 30-second epochs and a 5-class label space. Because ISRUC and HMC already provide two closely related EEG comparisons, we chose not to include a third similar EEG benchmark in the main table. Instead, we included Sleep-EDF (EOG) in the main results to assess performance on a different modality, which we felt was more informative for demonstrating generality beyond EEG. Sleep-EDF (EEG) was still used in the paper, but for the ablation studies rather than the main cross-model comparison. We will clarify this more explicitly in the revision, and if the reviewer considers it useful, we can also add the Sleep-EDF (EEG) results to the main comparison table.
>
> > **Editorial comments**
>
> We thank the reviewer for the editorial comments. We will make the necessary changes in the revised manuscript.
>
> **References**
>
> [1] Thévenaz P, Blu T, Unser M. Interpolation revisited [medical images application]. IEEE Trans Med Imaging. 2000;19(7):739-758.
>
> [2] Zhang R. Making convolutional networks shift-invariant again. In: Proceedings of the International Conference on Machine Learning. PMLR; 2019:7324-7334.

---

> > ### Author Rebuttal · Reviewer_uB3G · 2026-04-03
> >
> > Thank you for the response.
> >
> > The omission of Sleep-EDF (EEG) is reasonable, just ought to be made explicit. Other than that and the editorial comments, I have nothing more to add.

---

> > > ### Author Response · Authors · 2026-04-05
> > >
> > > We thank the reviewer again for their valuable comments and suggestions. We will make the omission of Sleep-EDF (EEG) explicit in the revised manuscript and will also incorporate the editorial improvements suggested by the reviewer.

---

### Official Review · Reviewer_fJsf · 2026-03-12

**Soundness:** 3
**Presentation:** 3
**Significance:** 2
**Originality:** 2
**Overall Recommendation:** 5
**Confidence:** 3

**Summary:**

Physiological signals (EEG/EOG/ECG) are measured on device-dependent sampling grids, and many pipelines rely on resampling/interpolation or fixed tokenization to make models “compatible” across acquisition settings. The paper argues this treats discretization as a prerequisite rather than an incidental measurement choice, motivating models whose computations are defined in physical time so they generalize across sampling rates. The technique uses a combination of SplineConv1d, Fourier projection pooling and Attention-based task head. Across multiple datasets NeurOCNN outperforms neural-operator baselines and is competitive with strong domain baselines. They evaluate zero-shot sampling-rate shift. Ablations support design choices.

**Compliance With Llm Reviewing Policy:**

Affirmed.

**Final Justification:**

My concerns were addressed.

**Key Questions For Authors:**

1. Can you report the inference time FLOPs, latency, and throughput for NeurOCNN and all baseline models, measured under the same hardware and evaluation settings?
2. How sensitive is the model to channel permutations? If the input channels are shuffled at test time, how much does performance change, and is the model designed to handle such permutations?
3. How does noise affect the model’s accuracy when the sampling resolution is also changed at test time? Please report performance across multiple noise levels and noise types, evaluated at test resolutions that were not used during training.

**Limitations:**

1. The paper does not study robustness to noise or missing data, even though both are common in real-world physiological time series and can significantly affect performance.
2. The paper does not report inference-time FLOPs, latency, or throughput for NeurOCNN or the baselines. Without these measurements, it is difficult to assess the model’s computational efficiency and practicality for deployment.

**Strengths And Weaknesses:**

**Strengths:**
1. Well-motivated problem setting.
2. Technique aligns with the claim of multiple resolutions.
3. Well controlled operator comparisons.
4. The paper performs useful ablations.
5. Proposed a compact model.


**Weakness:**
1. The paper does not report FLOPs, latency, or throughput for the discussed models. Parameter counts alone are not sufficient to quantify computational efficiency.
2. Missing values are a major challenge in real world time series acquisition. It is unclear whether the model can handle missing data and, if so, how performance changes as the missingness rate increases.
3. Noise robustness is mentioned as future work, but it is critical for physiological signals. The evaluation should include multiple noise types and quantify how noise affects performance, especially at test resolutions that were not used during training.
4. It is unclear whether the model is robust to permutations of input channels. If channels are shuffled at inference time, does performance remain stable?
4. When motivating the approach using PDEs, it would be important to emphasize that PDEs are often solved on irregular grids. In contrast, the proposed method assumes regular sampling, where the sampling rate is constant over time. This mismatch should be stated clearly, along with implications for irregularly sampled data.
5. The paper does not include visualizations of the datasets, even in an appendix. Simple plots would help illustrate signal complexity and support the motivation, particularly the claim about sudden jumps.

---

> ### Author Rebuttal · Authors · 2026-03-31
>
> We sincerely thank the reviewer for their thoughtful and constructive feedback. We address the reviewer’s main points below.
>
> > **Inference time FLOPS, latency, and throughput**
>
> We agree that efficiency should be assessed beyond parameter count alone, and we therefore report FLOPS, latency, and throughput under a shared evaluation setup.
>
> | Model | FLOPS | Latency (ms) | Throughput (samples/s) |
> | :--- | :--- | :--- | :--- |
> | NeurOCNN | 10.14M | 5.388 ± 0.031 | 2338.77 |
> | LaBraM | 743.58M | 2.553 ± 0.019 | 10322.47 |
> | AttnSleep | 256.11M | 1.098 ± 0.011 | 22244.46 |
> | EEGNet | 15.64M | 0.202 ± 0.004 | 32599.90 |
> | U-Sleep | 164.09M | 2.524 ± 0.023 | 14766.59 |
>
> While NeurOCNN is arithmetically compact, its reported latency should be interpreted in the context of an implementation focused on methodological validation rather than runtime optimization. We therefore view the measured latency as reflective of the present implementation rather than as a fundamental limit of the architecture. Crucially, our main claim is not minimal raw latency, but that NeurOCNN's compact, operator-based architecture delivers strong accuracy and discretization robustness.
>
> > **Sensitivity to channel permutations**
>
> Although NeurOCNN is not designed to be permutation-invariant, we performed an additional test to evaluate its sensitivity to channel-order perturbations. Specifically, on the ISRUC dataset, we randomly swapped the two input channels at test time with probability 0.5. Under this perturbation, the mean test accuracy averaged across all seven evaluation sampling rates decreased only from 77.76\% to 77.33\%, corresponding to a drop of 0.43 percentage points. This suggests that, although the model is not permutation-invariant by design, it exhibits limited sensitivity to mild channel-order perturbations.
>
> > **How does noise affect the model’s accuracy when the sampling resolution is also changed at test time?**
>
> We evaluated the proposed model on the ISRUC dataset under test-time sampling-rate ($f_s$) shift combined with additive noise corruption at unseen resolutions. Specifically, we considered powerline noise (PL) (50 Hz with first and second harmonics), baseline drift (BL) (0.05–0.5 Hz), EMG noise (20–60 Hz) [1], and white Gaussian noise (Gaussian), all at a signal-to-noise ratio of 5 dB, and evaluated all conditions at test sampling rates of 80, 200, 512, and 1024 Hz, while keeping training unchanged at the native training rate of 200 Hz.
>
> | Test $f_s$ | Model | Clean | PL | BL | EMG | Gaussian |
> | :--- | :--- | :--- | :--- | :--- | :--- | :--- |
> | 80 | NeurOCNN | 77.23 | **77.30 (+0.07)** | 72.69 (-4.55) | **77.26 (+0.02)** | **71.45 (-5.78)** |
> | | LaBraM | 62.00 | 56.70 (-5.31) | 59.85 (-2.15) | 20.57 (-41.43) | 28.95 (-33.06) |
> | | U-Sleep | 75.54 | 62.29 (-13.25) | **73.52 (-2.02)** | 43.09 (-32.45) | 47.73 (-27.81) |
> | 200 | NeurOCNN | 77.46 | **77.46 (0.00)** | 72.58 (-4.88) | **77.46 (0.00)** | **75.30 (-2.16)** |
> | | LaBraM | 64.32 | 57.39 (-6.93) | **63.52 (-0.80)** | 20.57 (-43.74) | 25.78 (-38.54) |
> | | U-Sleep | 79.46 | 67.15 (-12.31) | 77.21 (-2.25) | 43.84 (-35.62) | 48.92 (-30.53) |
> | 512 | NeurOCNN | 77.60 | **77.59 (-0.01)** | 72.66 (-4.94) | **77.57 (-0.03)** | **76.66 (-0.94)** |
> | | LaBraM | 64.54 | 57.78 (-6.75) | **63.54 (-1.00)** | 20.57 (-43.96) | 26.13 (-38.41) |
> | | U-Sleep | 79.32 | 66.61 (-12.71) | 77.44 (-1.88) | 43.69 (-35.63) | 48.88 (-30.44) |
> | 1024 | NeurOCNN | 77.64 | **77.64 (0.00)** | 72.70 (-4.94) | **77.60 (-0.04)** | **77.08 (-0.56)** |
> | | LaBraM | 64.36 | 57.62 (-6.74) | **63.68 (-0.68)** | 20.57 (-43.79) | 25.96 (-38.40) |
> | | U-Sleep | 79.51 | 66.92 (-12.58) | 77.36 (-2.14) | 43.80 (-35.71) | 48.90 (-30.61) |
>
> When exposed to common corruptions, NeurOCNN demonstrates significantly greater stability compared to both LaBraM and U-Sleep. Specifically, the model remains essentially unaffected by powerline and EMG noise, and exhibits only marginal degradation under baseline drift and Gaussian noise. Ultimately, these findings indicate that NeurOCNN offers far more consistent performance when subjected to combined distribution shifts involving both noise interference and resolution changes.
>
> > **Handling of missing values**
>
> We agree that missing data is an important real-world challenge. We did not include a controlled missingness study in the current submission and therefore do not want to overclaim on this point. At present, the model assumes fully observed, uniformly sampled windows after preprocessing. We view robustness to structured missingness or irregular dropout as an important direction for future work. We will state this more explicitly in the revised manuscript.
>
> > **Editorial comments**
>
> Thank you. We will make the necessary changes in the revised manuscript.
>
> **References**
>
> [1] Goncharova II, McFarland DJ, Vaughan TM, Wolpaw JR. EMG contamination of EEG: spectral and topographical characteristics. Clin Neurophysiol. 2003;114(9):1580-1593.

---

> > ### Author Rebuttal · Reviewer_fJsf · 2026-04-03
> >
> > My major concerns are addressed. I will update my score.

---

> > > ### Author Response · Authors · 2026-04-05
> > >
> > > We thank the reviewer again for their valuable comments and suggestions.

---

### Official Review · Reviewer_KXsL · 2026-03-19

**Soundness:** 3
**Presentation:** 3
**Significance:** 3
**Originality:** 3
**Overall Recommendation:** 4
**Confidence:** 3

**Summary:**

The paper introduces a new model architecture that is designed for time series analysis tasks, specifically targeting physiological domains. The model works by performing specially designed elements that perform continous time integrations and projections onto basis sets. The processed time series is then tokenized and put into a discrete transformer which performs the final task. The architecture is applied to a suite of benchmarks against other models designed for physiological time series. The paper also demonstrates its robustness to discretization

**Compliance With Llm Reviewing Policy:**

Affirmed.

**Final Justification:**

I think the performance of the different models with different sizes now makes sense, given the author's explanation. While the paper could have a broader scope by looking at a broader set of domains, I think it is a solid contribution.

**Key Questions For Authors:**

Clarifications:
- What are the trainable parameters in the splines and learnable kernels?
- How is the model trained / what are the loss functions? Is it only the cross entropy loss on the classification head, or is there a next-frame prediction loss too?
- Is there causal mask or a small window function in the integrals?
Minor:
- Table 3 - Is segment the window length or the time step size? Why is "8 s frames" written differently?
- The rightmost dots on Figure 2c and the lines on Figure 4 can't be differentiated.
- Stray "5" at the end of a paragraph on p2

**Limitations:**

The paper addresses limitations and broader impact.

**Strengths And Weaknesses:**

This is overall a strong paper that introduces are very robust way for processing time series. Verifying discretization robustness is an excellent thing to test; many methods that purport it do not actually satisfy it. Figure 5 looks a lot like the behavior studied in Ott, "ResNet After All? Neural ODEs and Their Numerical Solution", 2020 and Krishnapriyan, "Learning Continuous Models for Continuous Physics", 2023. The proposed spline method & the way that integration is handled does actually seem like it is passing the discretization invariance test.

One weakness of the presentation is that the proposed model is positioned very narrowly to only being applicable to one domain that is not of universal interest. I think the method could be easily applied to time series analysis problems (e.g. market data, plasma physics experiments), which would strengthen the paper.

The main issue I see in the paper is that the comparisons in Table 2 are not fair. The models have wildly different parameter counts. How is the comparison if the models are trained to be of similar sizes? Were all of the models trained for the point of this paper? How does NeurOCNN behave as it is scaled up or down? Why did you not make it larger to get the top accuracy in all datasets? Addressing this would be necessary for acceptance in the reviewer's opinion.

The presentation is overall good, but there are a few parts that are not explained; see the questions below.

---

> ### Author Rebuttal · Authors · 2026-03-31
>
> We sincerely thank the reviewer for their thoughtful and constructive feedback. We address the reviewer’s main points below.
>
> > **One weakness of the presentation is that the proposed model is positioned very narrowly to only being applicable to one domain that is not of universal interest.**
>
> We agree that the current positioning understates the generality of the method. NeurOCNN is not a physiology-specific architecture. It is a general operator-based approach for learning function-to-label mapping from continuous signals observed at varying discretizations or sampling rates. This problem extends well beyond physiological data in many time-series settings where robustness to acquisition resolution matters. We chose physiological signals as our evaluation domain because they are both practically important and especially challenging, making them a strong testbed rather than a restriction on the scope of the method. We will revise the paper to make this broader applicability more explicit.
>
> > **The comparisons in Table 2 are not fair. The models have wildly different parameter counts. How is the comparison if the models are trained to be of similar sizes? Were all of the models trained for the point of this paper? How does NeurOCNN behave as it is scaled up or down? Why did you not make it larger to get the top accuracy in all datasets?**
>
> We agree that parameter-count fairness is an important consideration. Our goal was not to claim that NeurOCNN wins through scale, rather, the model is intentionally compact, and our main claim is that its architectural inductive bias yields strong accuracy together with discretization robustness. In a depth-scaling experiment on the ISRUC dataset, under the same subject-independent train/test protocol used in the paper, increasing model size beyond the selected configuration did not improve held-out test accuracy: accuracy was 76.53\% at 135K parameters, 78.24\% at 319K parameters (used in the paper), 77.82\% at 566K parameters, 75.61\% at 1.30M parameters, and 74.97\% at 2.29M parameters. Thus, we did not increase the model size further because larger variants increased training accuracy but reduced test performance on unseen subjects, indicating overfitting rather than better generalization. Importantly, despite using substantially fewer parameters, NeurOCNN remains competitive with much larger domain-specific state-of-the-art baselines, supporting our claim that its gains arise from architectural design rather than scale. (These are single-run results, not 5-fold averages.)
>
> Regarding the reviewer’s comment on Table 2, this comparison appears in Table 1. All models in that table were trained and evaluated specifically for this paper under a common experimental protocol where all evaluated models are constrained to the architecture proposed by their authors. This is because, not all baselines can be resized in a comparable way. Specifically, LaBraM uses a released pretrained backbone, so its parameter count is inherited from that architecture. U-Sleep and AttnSleep were evaluated in their standard canonical forms as introduced by the original authors. Similarly, for the remaining baselines, we used the standard architectures/configurations commonly adopted in prior work, rather than re-designing each model to enforce parameter matching. We agree that this differs from a strictly parameter-matched comparison, and we will clarify this more explicitly in the revised manuscript.
>
> > **What are the trainable parameters in the splines and learnable kernels?**
>
> The trainable parameters in the spline convolution layer are the spline control-point values $\theta_{o,i} \in \mathbb{R}^P$ for each input-output channel pair, where $P$ is the number of control points. The knot locations are fixed, and the discrete convolution kernels are obtained by spline interpolation of these learned control points at the given sampling rate, so the kernel coefficients are not independent free parameters.
>
> > **How is the model trained / what are the loss functions? Is it only the cross entropy loss on the classification head, or is there a next-frame prediction loss too?**
>
> The model is trained end-to-end using AdamW with learning rate $10^{-3}$ and cross-entropy loss only. There is no next-frame prediction or auxiliary reconstruction objective.
>
> > **Is there causal mask or a small window function in the integrals? Minor:**
>
> The current model is not causal. It operates on fixed windows for classification, uses same-padded spline convolutions, and the token-mixing stage uses standard self-attention without a causal mask.
>
> > **Table 3 - Is segment the window length or the time step size? Why is "8 s frames" written differently?**
>
> Here, segment refers to the input epoch length in seconds, not the sampling step size. In the revision, we will explicitly mention this and remove "frames" for clarity.
>
> > **Editorial comments**
>
> Thank you. We will make the necessary changes in the revised manuscript.

---

> > ### Author Rebuttal · Reviewer_KXsL · 2026-04-06
> >
> > The authors have resolved my concerns with the fairness of the comparison -- the datasets may be limited and saturated in performance if scaling starts to overfit. The sweep of the NeurOCNN sizes you mention in your response should be mentioned in the paper, if it was not already. I have raised my score accordingly.

---

> > > ### Author Response · Authors · 2026-04-07
> > >
> > > We thank the reviewer again for their valuable comments and suggestions. We will include the NeurOCNN model-size sweep in the revised manuscript.

---

### Decision · Program_Chairs · 2026-04-30

**Decision:**

Accept (regular)

**Comment:**

The paper is focussed on time series analysis with an experimental evaluation on physiological time series. The issue being tackled is the fact that measurement considerations make discretization of a continuous signal imperative and discretization invariance (in the neural operator) sense is essential. To do so, the authors combine spline convolutions, fourier pooling and an attention mechanism on the latent tokens. Several experiments with neural operator baselines (FNO, U-NO etc) and domain specific baselines show encouraging results. The contribution is certainly novel, results impressive and underlying domain of great interest. Hence, the paper is at the level of acceptance in this venue. However, a camera ready version should certainly discuss the many limitations which were identified by the reviewers. These include i) noise sensitivity which authors have appropriately addressed during the rebuttal phase ii) the issue of missing data as well as irregularly sampled data iii) the possible broader application of the method and iv) the fact that the NO baselines are rather outdated. The issue of temporal continuity is being actively addressed in the operator learning community, with https://arxiv.org/pdf/2501.19205 being an example.